**Data Availability Statement:** All data from this publication are publicly available at Dryad doi:10.5061/dryad.pc866t1s4.

# Agricultural margins could enhance landscape connectivity for pollinating insects across the Central Valley of California, U.S.A.

**Thomas E. Dilts**[1]*, **Scott H. Black**[2], **Sarah M. Hoyle**[2], **Sarina J. Jepsen**[2], **Emily A. May**[2], **Matthew L. Forister**[3]

**1** Department of Natural Resources and Environmental Science, University of Nevada Reno, Reno, NV, United States of America, **2** Xerces Society for Invertebrate Conservation, Portland, OR, United States of America, **3** Program in Ecology, Evolution, and Conservation Biology, Department of Biology, University of Nevada Reno, Reno, NV, United States of America

* tdilts@unr.edu

## Abstract

One of the defining features of the Anthropocene is eroding ecosystem services, decreases in biodiversity, and overall reductions in the abundance of once-common organisms, including many insects that play innumerable roles in natural communities and agricultural systems that support human society. It is now clear that the preservation of insects cannot rely solely on the legal protection of natural areas far removed from the densest areas of human habitation. Instead, a critical challenge moving forward is to intelligently manage areas that include intensively farmed landscapes, such as the Central Valley of California. Here we attempt to meet this challenge with a tool for modeling landscape connectivity for insects (with pollinators in particular in mind) that builds on available information including lethality of pesticides and expert opinion on insect movement. Despite the massive fragmentation of the Central Valley, we find that connectivity is possible, especially utilizing the restoration or improvement of agricultural margins, which (in their summed area) exceed natural areas. Our modeling approach is flexible and can be used to address a wide range of questions regarding both changes in land cover as well as changes in pesticide application rates. Finally, we highlight key steps that could be taken moving forward and the great many knowledge gaps that could be addressed in the field to improve future iterations of our modeling approach.

## Introduction

Declines in insect abundance and diversity, which have been reported in recent years from around the world, pose a threat to the functioning of ecosystems and the stability of food chains supporting human society [1, 2]. Calls to action are many, but the challenges of insect conservation are profound [3, 4]. Agricultural areas have often suffered the most severe declines in insect abundance [5], yet these are precisely the areas where the need (from the human, economic perspective) for thriving insect populations is the most intense. There is also a mismatch between supply (wild bee abundance) and demand (cultivated area) for

**Funding:** M.L.F thanks the National Science Foundation (DEB-2114793). The funders had no role in study design, data collection and analysis, decision to publish, or preparation of the manuscript.

**Competing interests:** The authors have declared that no competing interests exist.

pollination in nearly half of the pollinator-dependent crop area in the United States, particularly in areas with significant acreage of highly pollinator-dependent crops such as almonds, blueberries, and apples [6]. The value of pollination services from wild pollinators to California agriculture is between $937 million and $2.4 billion per year [7].

The decline of pollinators in agriculture and elsewhere is driven by loss of habitat, degradation of remaining habitat by pesticide use and invasive species, along with pathogen infection and climate change [8, 9]. Loss and degradation of natural areas in California has been ongoing for over 100 years, and more than 260,000 acres of grassland and shrubland within California's Central Valley ecoregion (~3.7% of the land area) were either developed for housing or converted to agriculture between 1980 and 2000 [10]. Many farm properties in California contain little or no natural habitat, and when patches of pollinator habitat remain they tend to be isolated. Thus, any remaining pollinator diversity will be out of equilibrium and will in many cases include populations experiencing unsustainable levels of fragmentation [11]. Here we address the need for scientifically-driven management of the pollinator landscape using the Central Valley of California as a case study. The Central Valley is simultaneously an area of rapid human population growth, valuable agricultural land, and part of a global biodiversity hotspot (the California Floristic Province) that has seen steep declines in insect abundance and diversity over the last three decades [12, 13].

The management of lands for insect diversity faces particular challenges not faced by the management of other animal groups that are more easily observed or tracked and for which data on movement and dispersal are more likely to have been reported in the literature. With very few exceptions, the vast majority of insects are too small for long distance tracking devices, thus any information on movement across the landscape will be indirect at best [14]. Not only are behavioral and natural history data unavailable for most insects, land managers will often be in the position of wanting to maintain or rebuild insect communities that include unidentified or even undescribed species [15]. However, insect conservation can potentially benefit from the fact that small pieces of land that might have little value to larger animals or to the production of crops can still be suitable for insects. Small pieces of land adjacent to agricultural fields in particular offer great potential for enhancing pollinator connectivity as they are ubiquitous and often a simple, relatively linear shape that can facilitate restoration seeding and management. In aggregate, these habitats might comprise a sizable amount of habitat in many agro-ecosystems [16, 17] and may serve as stepping stones that allow movement among nearby larger or higher-quality lands. However, these ag-margin habitats also face a number of unique challenges. Because of their size, they can be particularly vulnerable to edge effects, including pesticide drift from adjacent agricultural fields which can negatively impact animals using the small patches [18]. In general, the weight of the unknown (e.g. scant or nonexistent data on pollinator demography and dispersal) has often overshadowed the potential opportunities presented by those marginal spaces.

To advance the use of marginal, agriculture-adjacent lands for conservation in our focal region, California's Central Valley, we have integrated a number of datasets focused on land use and pesticide application. We make a series of assumptions (detailed below) that let us investigate hypothetical insect movement across the valley and within a close-up study area that incorporates portions of the southern Sacramento Valley. Specifically, we address the following questions. (1) What does landscape connectivity look like from the perspective of flying insects, given the current state of agricultural, urban, and natural areas? (2) How would landscape connectivity be affected if urban and agricultural land types were characterized by greater or lesser resistance to movement for pollinators? (3) What would be the impact on connectivity if agricultural margins were restored or improved in ways that facilitated insect movement, or alternatively if agricultural production eliminated agricultural margins (such

that production encompassed all edges and open spaces near agricultural fields)? These questions represent a mix of applied and basic interests. In general (and from a basic perspective) we are interested in advancing our understanding of insect ecology in the Anthropocene, which involves understanding how fragmented and altered landscapes might be used by insects. From an applied perspective, answering the questions above has resulted in the creation of a tool that we hope will be used in the Central Valley to quantify the value of ecological restoration and help habitat restoration practitioners to identify the best (and worst) places to restore habitat for native bees and butterflies.

Although assumptions of our landscape model are discussed in greater detail below, we would like to highlight a central assumption here, and that is that our information on resistance to movement (also known as friction or impedance) comes mostly from published data on lethal doses associated with pesticides applied to different crop types. In other words, if a crop is typically treated with a high chemical $LD_{50}$ (i.e. a more lethal dose), we assume that fewer individual insects would be able to successfully move across a field of that crop relative to a field that is treated with a less toxic chemical; or, if they did move across the field, sublethal effects would impact subsequent function and fecundity [19–22]. We can depict resistance to movement on maps referred to as "resistance surfaces" that quantitatively express the relative difficulty of moving across one grid cell relative to another. We acknowledge that successful movement is a function of a great many things besides toxicity, and different insect species will respond in different ways to the same chemicals. This raises the question: is it worth building a model on such sweeping assumptions and simplifications? In answer to that question, we say: yes, because we have to start somewhere, and we believe that our results show a convincing robustness to at least some of our assumptions. Perhaps more important than the specific results reported here, it is our hope that the framework we put forward will inspire other researchers to collect more and better data to inform a new generation of landscape models that will support better management and conservation of insects in anthropogenic spaces.

## Materials and methods

### Study area

We delineated a study area that includes the agricultural Central Valley of California, encompassing the San Joaquin River (about the southern 2/3rds of the Central Valley) and Sacramento River (northern 1/3rd of the Central Valley) drainages as well as the Sacramento River Delta, and extending between 7 and 35 km into the adjacent foothills in order to ensure that sufficient natural or relatively undeveloped habitat adjacent to the agricultural matrix was included for the identification of start and endpoints associated with least-cost paths or corridors of potential insect movement. The study area is 68,005 km² encompassing portions of 20 counties, supports a population of over 7.2 million people, accounts for a $43 billion dollar agricultural economy [23], and 40% of the fruit, vegetable, and nut production in the United States [24]. Although the Central Valley itself is quite flat, it is bordered to the east by the Sierra Nevada foothills and to the west by the Coast Ranges. Both of these areas tend to have more natural or undeveloped lands and more grasslands than the Central Valley itself, although many of the common grasses at lower elevations surrounding the Central Valley are Eurasian exotics (*Hordeum* species, *Bromus* species, *Aegilops triuncialis*, *Ehrharta calycina*).

In addition to the larger Central Valley study area, we also investigate a 7,022 km² close-up study area that includes a swath of some of the richest and most agriculturally-productive lands in the northern Central Valley. Although largely agricultural, this study area contains three small metropolitan areas: Chico, Marysville/Yuba City, and Oroville/Thermalito and

numerous smaller agricultural centers such as those surrounding the towns of Williams, Colusa, and Gridley. The utility of the close-up area in our study was to assess how connectivity might differ depending upon the scale of the land being investigated and the density of source and destination points used in analyses. Examination of the close-up area also provides a more concrete example of how our landscape model could be used to target actual properties for preservation or restoration.

## Land use data

First, we combined a number of land cover products (LandIQ, C-CAP, NLCD) into a single unified land cover dataset. Next, we incorporated pesticide records collected by the California Department of Pesticide Regulation to generate maps of pesticide application rates across the Central Valley. The most important land cover layer used was the Crop Map developed by LandIQ for the California Department of Water Resources [25]. This layer was developed from 2014 National Agriculture Imagery Program aerial photographs at 1-meter resolution and ground data. The LandIQ layer is considered more accurate than the commonly-used USDA Cropscape layer because of the higher spatial resolution [26, 27]. We combined this layer with two other classified land cover maps for non-agricultural lands. The 2010 Coastal Change Analysis Program (C-CAP) is a system of land cover classification produced by the National Oceanic and Atmospheric Administration [28] that is available for coastal regions of the United States. This land cover map encompasses most of the southern portion of our study area. In areas not covered by C-CAP we used the USGS National Land Cover Classification of 2011 [29], which was primarily needed in the northern one-third of the Sacramento Valley and the lower two-thirds of the San Joaquin Valley.

Finally, three categories of urban lands ("urban impervious", "urban", and "urban greenspace") were assigned by calculating the Normalized Difference Vegetation Index (NDVI) from National Agriculture Imagery Program [30] data at a 5-meter resolution for urban areas only. Urban greenspace and urban impervious were separated from the standard urban class (mostly residential and small commercial buildings) by visually identifying appropriate breaks in NDVI using background imagery. Finally, we overlaid the California Protected Areas Database [31] and designated any conservation land "natural" regardless of what the land cover maps depicted. CPAD data did not include military lands, tribal lands, golf courses, or public lands not designated as open space.

## Resistance surfaces

The state of California maintains a database of annual pesticide applications to agricultural fields and makes these data available to the public in a database known as the Pesticide Information Portal [32]. We compiled data on moderately and highly bee-toxic pesticide application rates [33] for 2014, 2015, and 2016 for the top crop types in the study area as well as the urban land categories for each reported pesticide for 20 counties that comprised the majority of our study area. Our pipeline that translated application rates into landscape resistance is outlined in Fig 1 and also described here. First, we converted estimates from the Department of Pesticide Regulation in pounds (lbs) to micrograms for each crop. Honey bee (*Apis mellifera*) contact LD50 data (lethality for 50% of test subjects) were obtained from the EPA ECOTOX [34] and the US NIH PubChem [35] databases [36] and compiled for each pesticide. Using a process similar to that of Douglas et al. [36], micrograms applied were then divided by the LD50 (micrograms/bee) for each active ingredient to derive the total number of lethal doses applied to the crop group annually (an estimate of toxic load). This toxic load was then divided by the number of acres of that crop group grown in the Central Valley (determined

from 2014 land use imaging data) to provide a per-acre estimate of toxic loading for each pesticide. Per-acre estimates across individual pesticides were then summed for each crop to derive a total toxic load estimate for an acre of the crop type (in some cases, we treated similar crops as a group, such as the corn, sorghum and sudan group). DPR's Pesticide Use Reports do not include information on the planting of pesticide treated seed, which could lead to the underestimation of the toxic load for crops that use insecticide-treated seed including alfalfa, corn, cotton, rice, squash, sunflowers and wheat [37].

Urban pesticide use was estimated from DPR's Pesticide Use Reports for non-agricultural use categories in the 20 Central Valley counties. Since many urban pesticide applications are not required to be reported (including from residents applying pesticides around their homes), this method provides a potentially dramatic underestimate of actual rates of pesticide application to these landscapes. We did not include the 'structural pest control' category because it is not known which of these application methods (i.e. applications in and around buildings to control damage from termites and other structural pests) poses a significant risk to pollinators. Reported urban applications (in pounds) for each pesticide were converted to a per-acre toxic load using the same methodology as for crops described above.

The area of each crop type was derived from the LandIQ Crop Map [25] and urban area estimates came from the C-CAP and NLCD land cover maps. The Central Valley is of course a dynamic landscape, with changing crops and variation in pesticide applications from year to year even within single crop types. Although our model is temporally static, we wished to incorporate variation among years. To incorporate interannual variation in pesticide application rates we used each of the three years for which we had data (2014, 2015, and 2016) and assigned low, medium, and high pesticide application rates according to the lowest, median, and highest values within the three-year period.

As an example of the parameterization process (Fig 1) for one chemical in one year and for one land type, consider Spinosad (a commonly used insecticide in the US) application to urban lands. For the urban land class in 2014, 2608 lbs of Spinosad were applied in our study area. We converted that value to micrograms by multiplying 2608 by 453,592,370 (the total area) to yield 1,183,052,498,279 micrograms. The contact $LD_{50}$ for Spinosad is 0.0029 micrograms per bee. Dividing the micrograms of Spinosad applied by the contact $LD_{50}$ yields 407,949,137,337,493 lethal doses applied in that year (2014) across urban areas within the twenty Central Valley counties. Finally, to obtain our estimate of the lethal doses per acre from Spinosad for urban areas we divide the lethal doses by the acreage (552,724 urban acres) to yield 73,8069,624 lethal doses ($LD_{50s}$) per acre. Lethal doses per acre for different chemicals were added up for a given crop (or land) type.

To translate maps of pesticide applications rates into maps of resistance to movement for pollinators we used the results of a study by Jha and Kremen [38]. They compared genetic differentiation (as reflected in Fst values) among locations for the yellow-faced bumble bee (*Bombus vosnesenskii*) in eight urban landscapes in or near the Sacramento Valley in California. Among their conclusions, they estimate that urban lands with impervious surfaces were approximately eight times more resistant to movement than more natural land cover types. Specifically, Table 1 in Jha and Kremen [38] shows the most supported model has a resistance between 0.7 and 0.9 (model 4 –Table 1 in Jha and Kremen [38]) for urban land use categories (defined as having >20% impervious cover in Jha and Kremen [38]) compared to 0.1 for natural land cover types (the average of 0.7 and 0.9 is 0.8, which is then eight times more resistant than the natural land resistance of 0.1). In the absence of other or more general data, we based our estimate of resistance land cover types on this value (eight-fold difference from 0.1 to 0.8 between urban and natural) and extrapolated resistance to pesticide application rates in a linear manner. For example, we estimated 1,500,895,028 lethal doses (LD50s) per acre for the

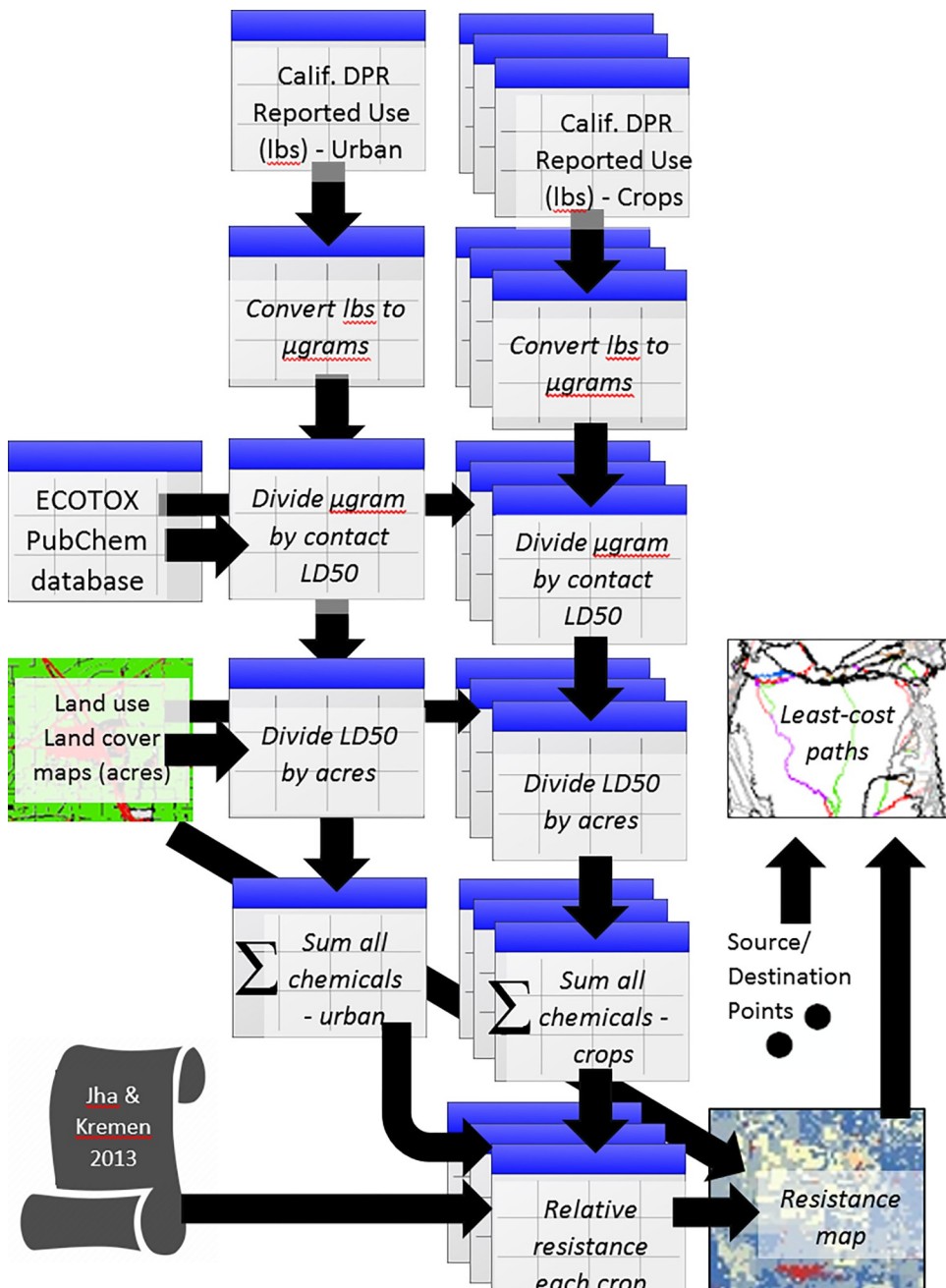

**Fig 1. Workflow used to generate the least-cost paths.** The five sources of input data include pesticide application reporting (State of California Department of Pesticide Regulation Pesticide Use Reporting database), toxicology databases (US Environmental Protection Agency ECOTOX database and the US National Institute of Health PubChem database), land cover maps (LandIQ crop map, NOAA Coastal Change Analysis Program, and US Geological Survey National Land Cover Map), estimates of landscape resistance for urban areas derived from Jha and Kremen (2013), and source and destination points created along the margins of the study area. Relative resistance of each crop/land cover type is shown in Table 1. Tabular data are depicted with the table icon whereas mapped data are shown using map icons. Italicized text indicates steps taken to process data. Non-italicized text is used to show input data.

**Table 1. Land cover types and resistance values assigned to each land cover / crop type under the low, medium, and high resistance scenarios (the table is sorted by the medium category).**

| Land Cover / Crop Type | Agriculture | % all CV | % interior | Low | Medium | High |
|---|---|---|---|---|---|---|
| Grassland | | 27.91 | 9.82 | 1 | 1 | 1 |
| Natural | | 11.95 | 3.87 | 1 | 1 | 1 |
| Rice | Y | 2.67 | 4.11 | 1 | 1 | 1 |
| Mixed Pasture | Y | 1.28 | 1.87 | 1 | 2 | 2 |
| Olives | Y | 0.29 | 0.42 | 1 | 2 | 2 |
| Sunflowers | Y | 0.29 | 0.45 | 1 | 2 | 2 |
| Carrots | Y | 0.20 | 0.31 | 2 | 4 | 6 |
| Corn, Sorghum and Sudan | Y | 3.63 | 5.59 | 4 | 5 | 6 |
| Plums, Prunes and Apricots | Y | 0.51 | 0.79 | 5 | 6 | 7 |
| Wheat | Y | 1.01 | 1.55 | 2 | 6 | 9 |
| Urban | | 7.67 | 8.74 | 7 | 8 | 9 |
| Pomegranates | Y | 0.19 | 0.30 | 6 | 12 | 17 |
| Beans (Dry) | Y | 0.24 | 0.36 | 12 | 14 | 16 |
| Agricultural Margin | | 6.15 | 8.98 | 7 | 15 | 20 |
| Urban Greenspace | | 0.27 | 0.30 | 7 | 15 | 20 |
| Potatoes and Sweet Potatoes | Y | 0.18 | 0.28 | 16 | 16 | 17 |
| Melons, Squash and Cucumbers | Y | 0.34 | 0.52 | 16 | 17 | 18 |
| Alfalfa and Alfalfa Mixtures | Y | 3.06 | 4.69 | 10 | 20 | 29 |
| Other | | 8.73 | 13.20 | 19 | 22 | 25 |
| Grapes | Y | 3.76 | 5.75 | 22 | 24 | 26 |
| Safflower | Y | 0.24 | 0.36 | 21 | 26 | 31 |
| Tomatoes | Y | 1.74 | 2.67 | 28 | 29 | 30 |
| Almonds | Y | 6.69 | 10.30 | 27 | 32 | 37 |
| Cherries | Y | 0.25 | 0.39 | 32 | 33 | 34 |
| Onions and Garlic | Y | 0.23 | 0.35 | 30 | 34 | 39 |
| Pistachios | Y | 2.01 | 3.08 | 30 | 34 | 39 |
| Peaches/Nectarines | Y | 0.47 | 0.72 | 34 | 35 | 36 |
| Walnuts | Y | 2.13 | 3.27 | 35 | 38 | 40 |
| Bare Soil | | 1.09 | 0.60 | 20 | 40 | 100 |
| Urban Impervious | | 0.89 | 1.25 | 10 | 40 | 100 |
| Cotton | Y | 1.29 | 1.99 | 42 | 45 | 48 |
| Citrus | Y | 1.39 | 2.13 | 87 | 92 | 98 |
| Water | | 1.24 | 0.99 | 100 | 100 | 100 |

"Y" in the agriculture category indicates that the land cover type was derived from the LandIQ crop type map. The remaining land cover categories were obtained from NOAA C-CAP when available or NLCD 2011 data when C-CAP was not available. "% all CV" indicates the percentage of the Central Valley study area in that land cover, and "% interior" indicates percentage in the Central Valley study area.

urban land cover class (see previous section). Pomegranates had an average of 2,220,400,221 lethal doses per acre, and are thus 1.47 times (2,220,400,221 / 1,500,895,028) more toxic than urban areas; we can then calculate a resistance estimate for pomegranate fields as 11.76 (1.47 x 8), rounding to 12 to produce a whole number (in other words, 12 times the resistance of the baseline natural areas given a value of 1).

Resistance values shown in Table 1 were mostly determined with the protocol described above but in some cases were refined based on expert opinion since resistance to movement is not only related to pesticide application rates. For example, small-bodied insects may in some

cases be unable to successfully navigate across large stretches of open water [39]. The land cover types for which resistance was estimated or modified using expert opinion resulted in a wide spread between high and low values reflecting a high level of uncertainty. The following land cover types were estimated solely based on expert opinion: agricultural margin, urban greenspace, bare soil, and open water. It is important to note that expert-assigned resistance for agricultural margins was not influenced by the neighboring crop type; instead, we imagined a typical weedy edge (sometimes but of course not always containing nectar and even exotic host plants for some herbivorous insects like butterflies) with mid-range resistance values (Table 1), but this varies in our experimental manipulations, described below. Resistance for "other" as a category that encompassed unidentified land types or types that were especially rare was estimated using the average of all crop types.

## Resistance scenarios and experiments

To understand the extent to which uncertainty in relative resistance values for different land types potentially affects connectivity across the landscape, we examined least-cost paths and associated indices (described in detail below) with resistance surfaces corresponding to the low, medium, and high values shown in Table 1. In addition, we ran two experiments in which all agricultural margins were first restored (improved) to natural habitat, simulating the potential effects that pollinator hedgerows or restoration of native vegetation could have on habitat connectivity; and, secondly, agricultural fields were extended in space such that margins were eliminated. Thus, in total, we had a three-by-three design (with nine total levels) in which the low, medium, and high resistance values were crossed with the margin experiment consisting of three levels: restored margins, status quo, and eliminated margins.

## Analyses: Connectivity models

Corridor mapping was performed using least-cost paths [40] connecting source points on the western and eastern perimeter of the study area. Least-cost paths were calculated using the Cost Distance and Cost Path as Polyline tools in ArcGIS Spatial Analyst version 10.7.1. Cost-distance was calculated using Dijkstra's algorithm [41], which is an iterative algorithm that finds the shortest distance between all nodes and a source node in a graph, and the least-cost path is identified by taking the accumulated cost (resistance) traversed. The average distance between a point and its nearest neighbor was about 5 km, resulting in 205 source points along the western perimeter and 205 source points along the eastern perimeter. For each source point on the western perimeter we calculated the least-cost path to all points in the eastern perimeter within a 125 km Euclidean distance buffer, which resulted in a variable number of paths but typically around 36 paths per point. To account for variability in start and end points we repeated the process totaling 20 iterations using random points within a 5 km buffer of original source and end points. We do not necessarily expect any individual insect to traverse the full length of these least-cost paths. Rather these paths are designed to represent paths of lowest resistance that could be traversed over the course of many generations. Least-cost paths were calculated using each of the three resistance surfaces (low, medium, and high) crossed with each of the three agricultural margins scenarios (current-day, enhanced, and converted to agriculture) in a factorial manner resulting in a total of nine scenarios. Finally, to identify corridors through the agricultural matrix of the Central Valley rather than along the edge of the valley, we created a refined study area that only encompassed the primary agricultural and urban areas in the Central Valley (Fig 2).

We applied the same least-cost modeling process that was used on the entire Central Valley to the close-up study area, and used the same 30-meter resistance surfaces in the same factorial

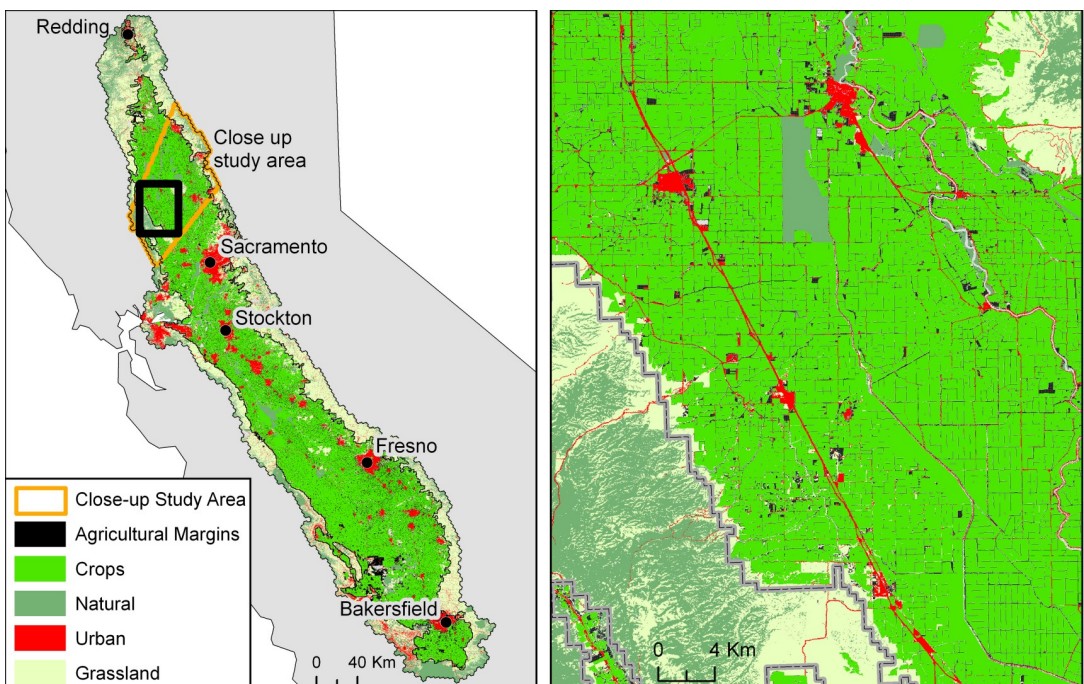

**Fig 2. Major land cover types in the Central Valley study area.** All 26 crop types have been combined for the purpose of map legibility. The agricultural margins appear as tiny dots scattered throughout the Central Valley (left panel), but appear much more clearly as linear features in close-up map (right panel). There is a black line around the total study area (Central Valley and adjoining lands; see text for details) as well as the predominantly agricultural inner study area, which is also shown with the gray line in the right panel. The darker green areas are conservation lands that are treated as natural in our analyses.

combination (3 levels of resistance x 3 agricultural margins scenarios). In the close-up study area we used 43 origin points located along the western perimeter of the Central Valley spaced at 1.8 km, rather than the 5 km spacing used in the Central Valley study area. We used 80 destination points located along the eastern perimeter of the portion of the Central Valley encompassed by the smaller study area.

## Results

### Land cover analysis

Within the Central Valley of California we estimate from land cover maps that 4,182 $km^2$ are agricultural margins, which constitute 6.15% of the study area and cumulatively are larger than any one crop type, with the exception of almonds (Table 1; Fig 2A). Unlike most crops, agricultural margins are distributed throughout the Central Valley, and typically occupy narrow linear patches (Fig 2B). If we only consider the inner study area (predominantly agricultural lands as shown by the thin black line in Fig 2A) the proportion of the area occupied by agricultural margins is 8.89%.

Crops accounted for 35.26% of the larger Central Valley that includes both the inner and outer study areas (Fig 2). Within the more intensively cropped inner study area crops accounted for 65.46% of the total area. Urban land cover types accounted for 8.83% and 10.29% in the larger and in the inner study areas, respectively. Natural areas occupied 11.95% of the larger study area, but only 3.86% of the inner study area, reflecting the fact that natural habitat is more abundant in foothill areas than in areas dominated by intensive croplands. The remaining land cover was grassland constituting 37.81% of the larger study area or 11.50% of

the inner study area. Within the inner area, most natural areas consist of federal, state, and locally protected lands, such as parks and wildlife refuges. Within the inner area, agricultural margins occupied an area 2.3 times greater than natural areas.

## Resistance values

Average resistance values among crop types were 19.4, 22.0, and 24.5 for the low, moderate, and high resistance scenarios. The standard deviations among crop types for the three scenarios were 19.6, 20.2, and 21.4 reflecting the wide range of resistance values (Table 1). The crop with the lowest resistance-to-movement was rice due to its very low reported pesticides application rates. However, rice can be planted with insecticide-treated seed, which is not reflected in our resistance estimates [37]. Citrus had consistently high resistance values of 87, 92, and 98 reflecting very high reported $LD_{50}$ values. Among the crop types with high resistance, citrus was followed by walnuts (35, 38, 40), peaches/nectarines (34, 35, 36), pistachios (30, 34, 39), onions and garlic (30, 34, 39), cherries (32, 33, 34), almonds (27, 32, 37), tomatoes (28, 29, 30), safflower (21, 26, 31), and grapes (22, 24, 26), with each number representing the resistance to movement used for the low, moderate, and high resistance scenarios (generated with the pipeline described in Fig 1).

The spatial arrangement of resistance to movement also varies dramatically through the Central Valley, with foothill regions around the perimeter of the valley having relatively low resistance to movement (Fig 3). Differences among crop types are apparent on the map (Fig 3) with high resistance crops, such as citrus, showing up for example in the southern San Joaquin Valley (southernmost 1/3$^{rd}$ of the study area). One difference among the resistance scenarios (low, medium and high, from Table 1) is that urban areas varied dramatically in their resistance to movement as indicated by the nearly blue color on the map for the low resistance scenario, to the red color for the high resistance scenario (see the Sacramento area in particular, Fig 3). The larger differences among the low, medium, and high resistance scenarios for the urban classes reflect the fact that expert opinion was used to select these resistance levels and uncertainty is high.

## Paths across the Central Valley

Least-cost paths were more common around the perimeter of the Central Valley than in the area of intensive agriculture (inner study area) due to the presence of low-resistance land cover types, such as natural or undeveloped lands and grasslands (Fig 4). In contrast, the number of paths in the inner study area were relatively few (around 14 in the current margins scenario) and tended to be concentrated along low-resistance routes. The number and density of paths crossing the Sacramento Valley (number of paths = 13) was far greater than across the San Joaquin Valley (number of paths = 7) despite the fact that the San Joaquin is longer (434 km vs. 284 km straight line). The majority of least-cost paths were common among all of the resistance scenarios. For example, in the current agricultural margins scenario 85.0% of path lengths were common among all three resistance levels, and 6.6% were common among two resistance scenarios. Among the single resistance levels, low resistance had more unique paths (7.1%) compared to moderate (0.7%) or high (0.4%). The restored margins scenario had a similar proportion of paths, which coincided among all three resistance levels at 85.1%. The number of paths was much greater, however, than under the current (or status quo) agricultural margins scenario, with the number of paths being greater in both the Sacramento and San Joaquin Valleys. The no margins scenario tended to have a similar total path length as the current scenario (3,783 vs. 3,675 km) in that 88% of pathways were located in the same areas as the current margins scenario. An interactive map of connectivity routes is available at https://arcg.is/1eGvDv.

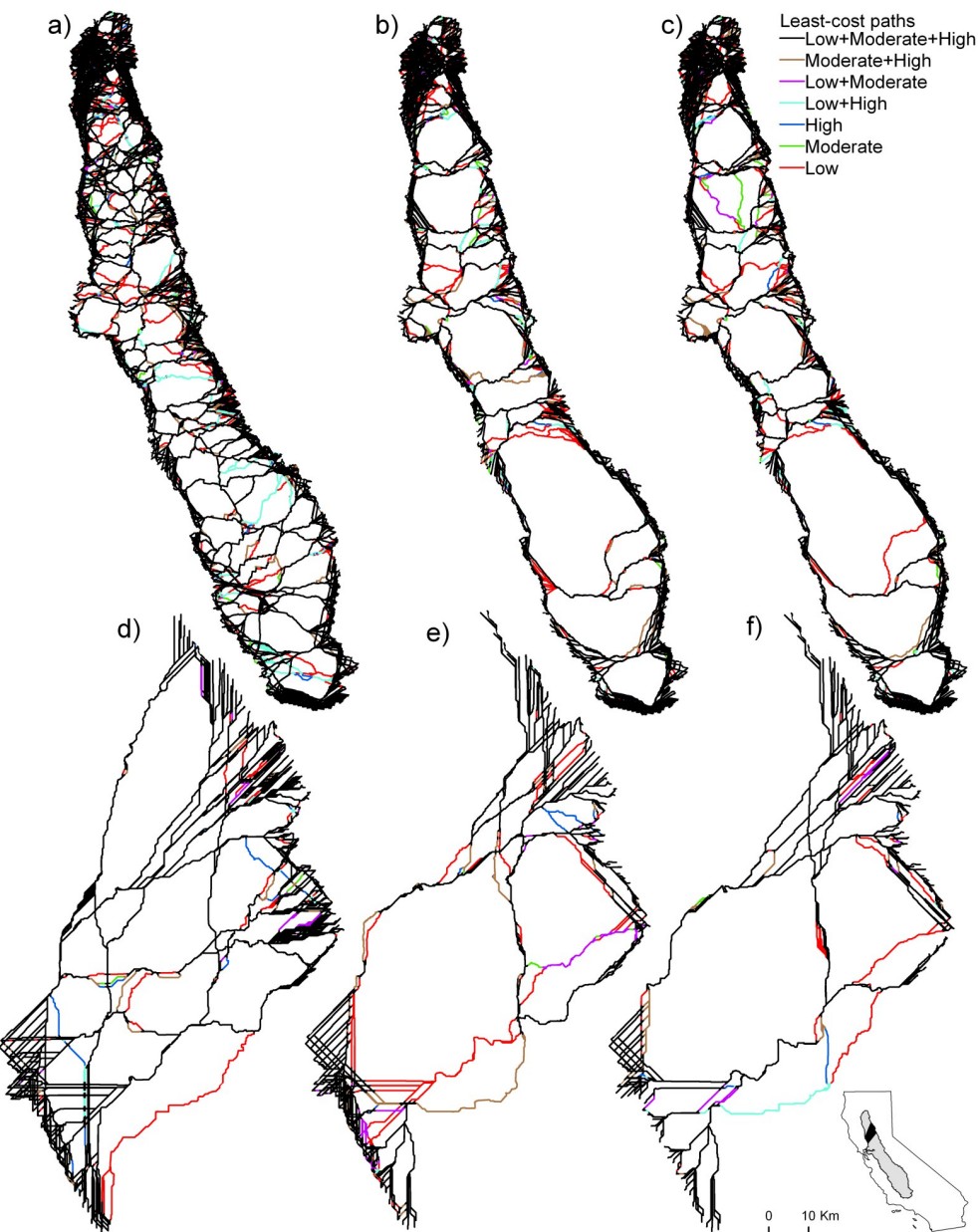

**Fig 3. Resistance to movement maps used to parameterize the Central Valley pollinator connectivity models.**
Maps depict the resistance surfaces used for the a) restored margins scenario, b) current margins scenario, and c) no margins scenario. Colors range from low resistance (blue) to high resistance (red). A resistance value of 1 is equivalent to isolation-by-distance, while a value of 100 indicates that a grid cell is 100 times more difficult to traverse than the lowest-resistance cell.

## Agricultural margins and movement across the valley

The restored agricultural margins scenario included the largest number of paths, which were well-distributed across both the Sacramento and San Joaquin River Valleys (Fig 4A). Total path length, including both the portions inside and outside of the inner buffer, provide a measure of the directness of routes across the Central Valley. The restored margins scenario had an average of 1.59 million km, whereas the current and no margins scenarios had an average

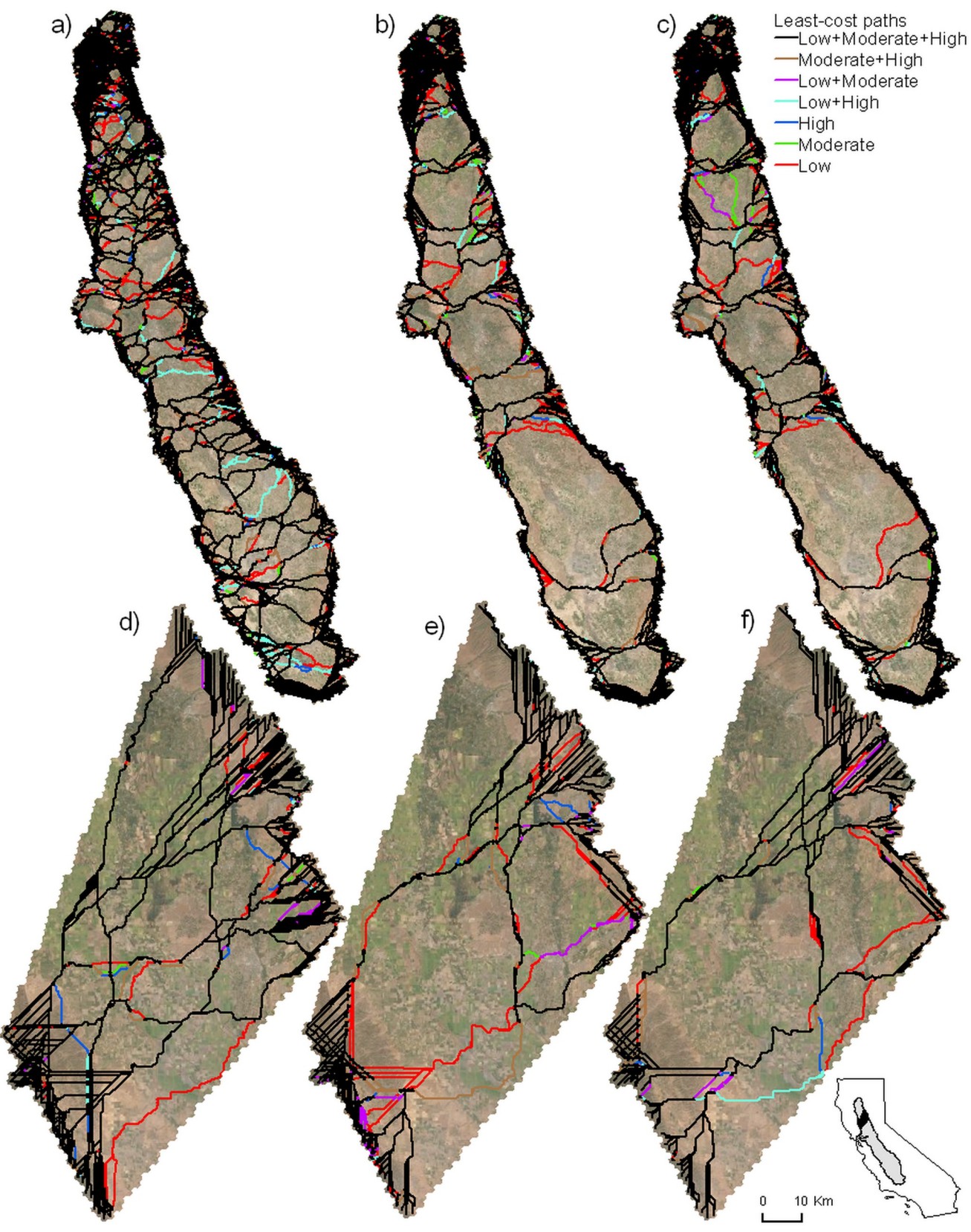

**Fig 4. Least-cost paths across the Central Valley and the close-up study area.** Line colors depict all possible combinations of resistance levels (low, moderate, high, low+moderate, etc.) for the a) restored margins scenario, b) current margins scenario, and c) no margins scenario across the Central Valley. The same line colors are used to depict the same combinations of resistance for the d) restored margins scenario, e) current margins scenario, and f) no margins scenario for the close-up study area. An interactive version of all scenarios can be found at: https://arcg.is/1eGvDv.

of 1.86 million km and 1.87 million km, respectively (Fig 5A). This indicates that the lowest cost routes (for the current and no margins scenarios) are both longer and result in higher movement costs compared to the restored margins scenario. Another metric that measures the redundancy in least-cost paths is the unique path length within the inner study area (Fig 5B). The restored scenario had an average length of 10,613 km compared to 3,883 km and 4,168 km for the current and no margins scenarios. This suggests that there is about 2.6 times the combined length of unique non-overlapping least-cost paths in the inner study area in the restored scenario compared to the other two scenarios.

The average distance to the nearest least-cost path provides an indication of how difficult it might be for individuals to access a least-cost path. Grid cells in the restored scenario were an average of 3.34 km from a least-cost path, whereas in the current scenario they were 10.42 km from the nearest path (Fig 5C). In the no margins scenario, they were 11.28 km, on average, from the nearest path. The average number of convergent paths indicates the average number of least-cost paths traversing a single cell within the inner area considering only those cells that are located within paths. The average number of convergent paths was 86, 193, and 180 for the restored, current, and no margins scenarios, respectively (Fig 5D). These metrics are all largely congruent suggesting that the restored scenario provides a greater abundance and diversity of pathways for pollinators compared to both the current and no margins scenarios. Differences among the current and no margins scenarios were far less, and in the case of the average number of convergent paths slightly more paths converged in the current scenario compared to the no margins scenarios. The other three metrics suggested slightly greater connectivity for the current scenario over the no margins scenario although these differences were about an order of magnitude lower than the differences between current and restored.

The types of land cover categories traversed by the least cost paths also differed when we experimentally restored or removed agricultural margins (Fig 6A). Agricultural margins constitute 31.3% of path length in the restored scenario compared to 1.7% and 4.3% for the current and no margins scenarios. Natural areas proportionally account for more of the path length in the current and no margins scenario: 26.4% and 25.5% compared to 17.0% for restored. However, the greater proportion of natural areas within the least-cost paths is also accompanied by increased path length (Fig 5A) suggesting that pollinators may have to traverse a greater Euclidean distance as well as greater resistance-to-movement in order to access those natural areas. Furthermore, the proportion of path lengths consisting of urban, cropland, and grassland is greater for the current and no margins scenarios compared to the restored scenario.

## Comparing uncertainty in resistance with manipulation of margins

Resistance scenarios (low, medium and high values from Table 1) were far less impactful on results than the agricultural margin scenarios, which can be seen by comparing differences among low, medium and high results with differences among restored, current and no margins scenarios in Fig 5. Total path length averaged 1.73 million km, 1.78 million km, and 1.87 million km for low, moderate, and high resistance, respectively, when averaged across the agricultural margins scenarios. Unique path lengths were 6992, 5977, and 5695 km for low, moderate, and high levels of resistance. The average distance to the nearest path was 7.77, 8.55, and 8.71 km and the average number of convergent paths was 139, 156, and 163 for low, moderate,

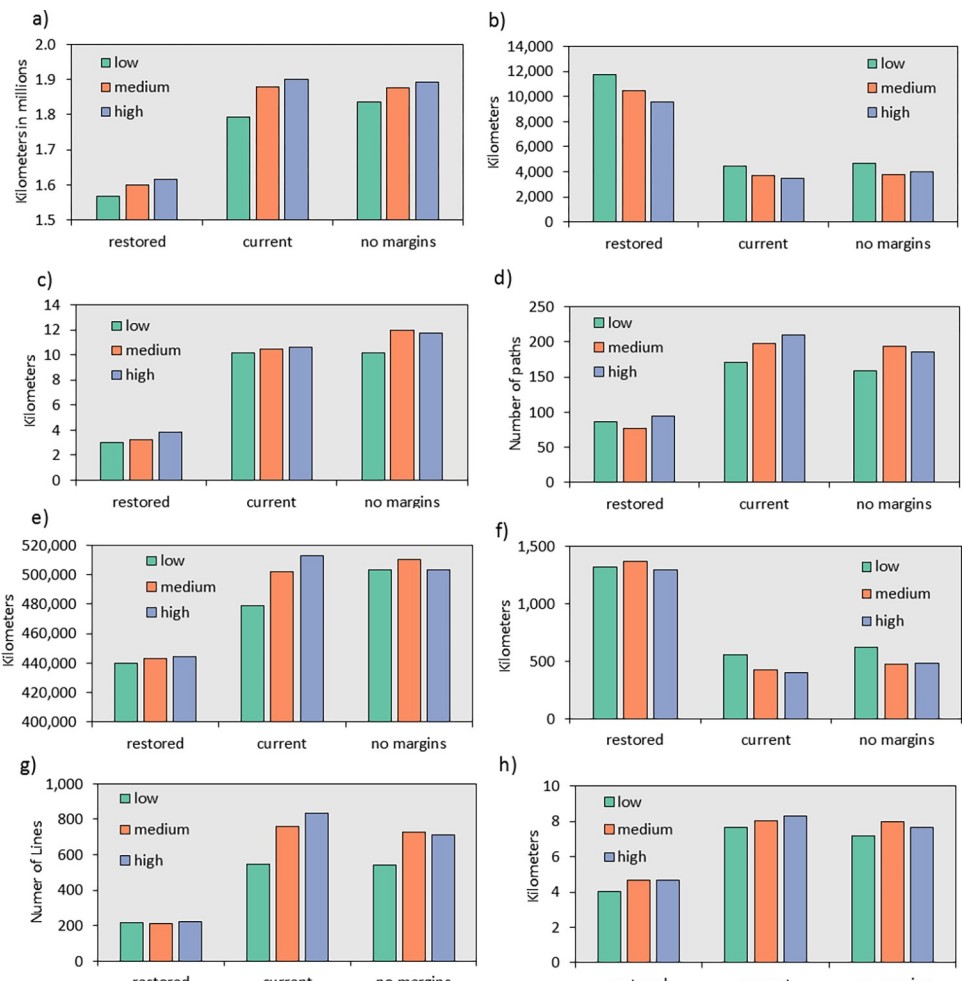

**Fig 5. Summary statistics describing least-cost paths across the Central Valley and the close-up study area.** a) Total path length of least-cost paths connecting the western and eastern sides of the Central Valley including the outer area, b) unique path length in the inner area for the Central Valley, c) average distance to the nearest path in the inner area for the Central Valley, d) average number of paths in the inner area for the Central Valley, e) total path length of least-cost paths connecting the western and eastern sides of the close-up study area including the outer area, f) unique path length in the inner area for the close up study area, g) average distance to the nearest path in the inner area for the close up study area, and h) average number of paths in the inner area for the close-up study area. Note the differences in the scale of the y-axis among graphs.

and high resistance scenarios. These differences are about an order of magnitude less than the differences between the restored and current agricultural margins scenarios. Furthermore, the differences among the mapped least-cost paths differ less between resistance scenarios than they differ among agricultural margin scenarios (Fig 4). The proportions of different land cover types were similar within agricultural margin scenarios, but differed more between resistance scenarios (Fig 4).

## A closer look at a sub-section of the Central Valley

In general, differences in properties of the least-cost paths among the scenarios using the close-up study area resembled those for the broader Central Valley study area. As with the Central Valley study area, the scenario with restored margins had the greatest number of least-

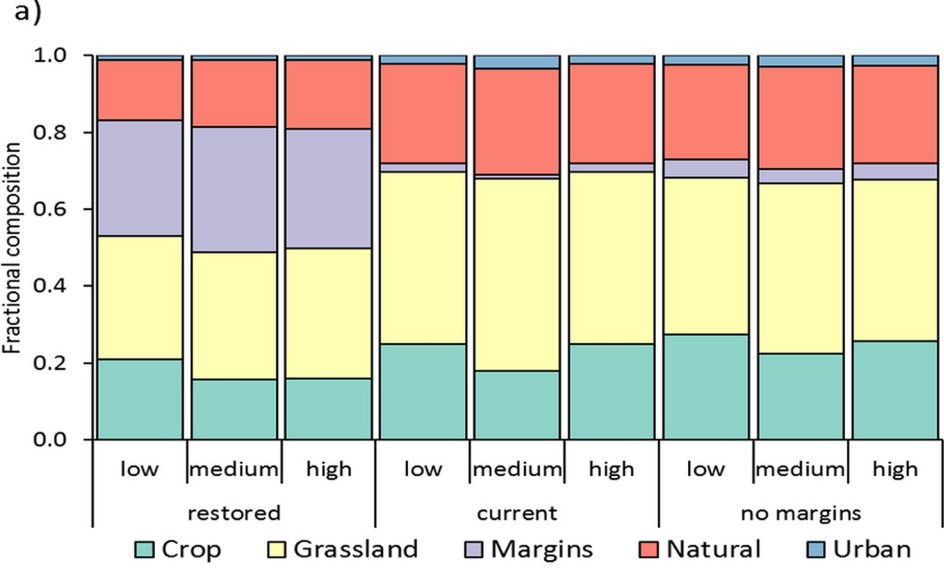

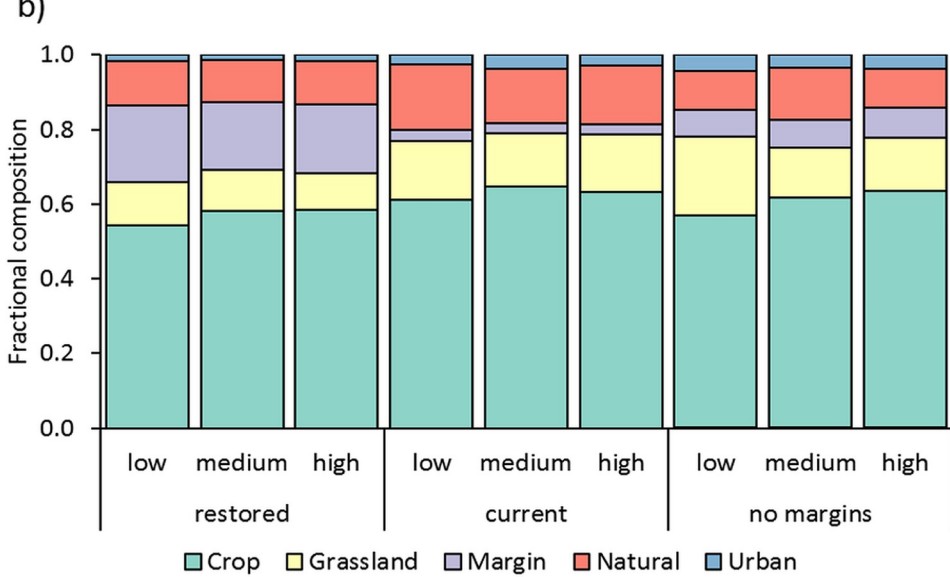

**Fig 6. Land cover classes traversed by least-cost paths for each of the three agricultural margin scenarios crossed with the three resistance scenarios.** a) Land cover classes for the Central Valley study area and b) the close-up study area. All of the crops are condensed into a single category, and the three urban categories have been condensed into a single category.

cost paths crossing the area (Fig 4A), while the current and no margins scenarios had far fewer paths and differed from one another much less than from the restored scenario. The restored scenario also had a higher proportion of paths that were shared across all resistance scenarios compared to the other two margin scenarios (reduced or status quo). Our four metrics support the pattern seen on the maps in which the restored-margins scenario differed the most strongly from the other two scenarios (Fig 5). Total path length increased dramatically between the restored scenario and the other two scenarios suggesting longer and more circuitous paths due to higher levels of landscape resistance (Fig 5E). Furthermore, the current

scenario differed most across different resistance levels, whereas both the natural and no margins scenarios showed little difference depending upon the resistance. The number of unique paths (Fig 5F) also showed a pattern consistent with what can be seen on the maps: the restored-margins scenario differed strongly from the other two scenarios showing more pathways compared to current and no margins scenarios. The average distance to a least-cost path was approximately double for the current and no margins scenarios relative to the restored scenario (Fig 5G), and the average number of paths that converged was approximately three times greater for the current and no margins scenarios compared to the restored scenarios (Fig 5H). Least-cost paths were far more likely to incorporate agricultural margins in the restored scenario compared to the current and no margins scenarios (Fig 6).

## Discussion

Insects are essential to functioning terrestrial ecosystems and to human society, yet the great importance and diversity of insects are rivalled in magnitude by how little we understand the habits and life histories of most individual species [15]. Not surprisingly, much of the history of conservation and management has focused on larger and more easily-observed insects [42], for example a few charismatic butterflies or bumble bees, which has left us with critical knowledge gaps as we plan a response to the crisis of insect declines [43–46]. Here we have taken a small step towards addressing this need by creating a modeling framework in which resistance, informed mostly by pesticide lethality, can be visualized and explored across one of the most important and diverse agricultural landscapes in the United States. In addition to the creation of that modeling framework, which we hope will be improved upon by other researchers, we have been motivated in particular by the desire to evaluate the potential importance of agriculture-adjacent, marginal spaces.

Agricultural margins have been a focus of conservation in the Central Valley for decades [47]. More recently conservationists have started to plant multi-storied woody hedgerows with herbaceous understories thereby maximizing plant and flower diversity while minimizing the space needed for conservation across the farm landscape [48, 49]. Field edge habitat in homogeneous agricultural landscapes can serve multiple purposes including enhanced biodiversity, water quality protection, and habitat for beneficial insects, such as native bees and natural enemies [44]. Hedgerows are economically viable to growers because they enhance populations of insects that are beneficial for pest control and pollination, and USDA cost share programs allow for a quicker return on a hedgerow investment [47]. Our paper shows that linear habitats could provide substantial areas for habitat restoration and could also help connect habitats across the Central Valley as we found that drainage ditches, field edges and uncultivated borders all together cover more land in the Central Valley of California than any one crop type. Beyond the identification of specific areas that might be usefully targeted for restoration, our primary finding is that these small linear spaces are a mostly-untapped resource for the conservation and management of insects, including beneficial pollinators, in the region. Our work also demonstrates the feasibility of restoring habitat connectivity across the Central Valley using only the restoration of marginal agricultural spaces. Maintaining and increasing habitat and habitat connections will enable insects to move among locations, increasing gene flow and preventing populations from becoming too small, which in turn could help insects shift or expand their distributions in response to a changing climate. Of course, not all species will change geographic distributions in response to warming or drying regional conditions, but increasing habitat connectivity will provide the opportunity for those that can, and might improve the adaptability (through larger population sizes) of less mobile species. Creating habitat connections across agricultural landscapes may help pollinators reach important refugia,

and may provide important microhabitats for pollinators and other insects to use. The benefits of connectivity include more stable pollinator communities and sustainable pollination services.

## What does landscape connectivity look like from the perspective of flying insects, given the current state of agricultural, urban, and natural areas?

In its current state, the Central Valley of California is a complex, heterogeneous and dramatically fragmented landscape but there are some important differences among regions within the valley. Resistance to movement was estimated to be about 4 to 5 times higher in the inner study area (a region of intensive cropping) compared to outer margins on the Central Valley. Maps of least-cost paths show higher resistance for cross-valley movements in the San Joaquin Valley compared to Sacramento Valley, which is, at least partially, driven by the presence of crops farther south with higher pesticide use–such as citrus–relative to crops with lower pesticide use–such as rice. Land cover types mapped as natural (which are largely designated conservation lands, like state and federal refuges and county and city parks) are a small fraction of the total land base within the inner study area at 3.8%. In contrast, agricultural margins make up an area more than double the area of all natural lands put together (within the inner Central Valley, the area occupied by agricultural margins is 8.9%, compared to the 3.8% for natural areas). These marginal lands have the potential to provide important habitat that facilitates connectivity.

We are not the first to model connectivity in the Central Valley, although our study may be the first to model connectivity with a focus on insects and their habitats. The California Essential Habitat Connectivity Project [50] convened a large number of experts and derived a single resistance surface using expert opinion. They then connected natural landscape blocks above a minimum size (< 2,000 acres in the Central Valley). The resulting corridor map for the Central Valley (page 53 in the California Essential Habitat Connectivity Project) only reveals two corridors crossing the Sacramento Valley, both near the northernmost extent of our study area. In addition, Spencer et al. [50] show a number of linear reaches of habitat extending out into the valley to connect blocks of natural lands within the valley to the adjacent foothills. One extends from the west side of the valley to the Sutter Buttes. Another extends from the Sierra Nevada parallel to the Mokelumne River and approximates our Corridor 6. There are several notable differences in the methodology between our study and Spencer et al. [50], which considered connectivity among protected habitat blocks. In contrast, our study includes many smaller pieces of habitat including agricultural margins. Another key difference between our approach and the approach of Spencer et al. [50] is that they used expert knowledge to parameterize a resistance surface. Our study, in contrast, uses empirical data on pesticide application rates to parameterize a resistance surface with some expert knowledge for land cover classes in which little information is available. To our knowledge, this use of pesticide information in a landscape-movement context is novel. Our approach may be less applicable to taxa that are less sensitive to pesticides than insect pollinators, although could still be relevant for animals that depend on insects such bats and insectivorous birds.

Huber et al. [51] created corridors for eight umbrella species (tule elk, bobcat, giant garter snake, pronghorn, western yellow-billed cuckoo, riparian forest, San Joaquin kit fox, Swainson's hawk) across the Central Valley. They used a combination of habitat models with "restorability" criteria, such as road density and land use, and calculated a density surface for each focal species. They used the Marxan software to optimize the spatial placement of hypothetical nature reserves and then identified least-cost corridors in GIS using a resistance surface that was the inverse of the habitat models for each species. Their final map shows a much larger

number of corridors than Spencer et al. [50] and superficially bears more resemblance to our corridor map, although it would take overlaying corridors in GIS to determine exactly how much overlap actually exists (an exercise beyond the scope of the current project). Choe et al. [52] examined the maps from three different studies using four different connectivity modeling approaches across the state of California, including the Central Valley. The four different approaches included the Metacorridor approach [52], Network Flow Analysis [53], Land Facets analysis [54], and Omniscape [55]. They found that there was relatively little overlap between the four corridor mapping studies. Each study used a different methodological approach, differed in their research objectives, and differed in the type of resistance surfaces used in models. Two approaches used focal plant species (Metacorridors and Network Flow Analysis) whereas the other two connected areas of low human impact (Land Facet Corridors and Omniscape). Importantly, none of the four studies reviewed by Choe et al. [52] include insect pollinators as a focal species, and maps from all four studies (as shown in Choe et al. [52], Fig 1) show a lack of corridors in the Central Valley. Corridor maps that omit highly developed landscapes may give the impression that highly modified landscapes are not restorable or valuable for connectivity. In contrast, changing economic and social conditions may present opportunities for habitat enhancement in highly modified landscapes that even exceed those of natural landscapes. For example, adoption of agricultural practices that use no or very low levels of pesticides may greatly enhance connectivity. Losses of farmland due to drought may also present opportunities for habitat enhancement by incorporating plantings of low-water plants that provide resources for pollinators.

Agricultural margins have nearly been uniformly ignored in habitat connectivity studies up until now, largely because they are small, and high resolution maps of agricultural margins are frequently not available. Despite being omitted from most habitat connectivity modeling projects, academic researchers and restoration practitioners have long called for more attention to agricultural margins [56, 57]. As stated above, our research suggests that agricultural margins can be critically important for increasing connectivity: agricultural margins constitute an area more than two times greater than designated conservation lands in the Central Valley. Moreover, the potential importance of agricultural margins is evident even in the context of the considerable uncertainty that is represented by our low, medium and high resistance scenarios.

## What would be the impact on connectivity if agricultural margins were improved for insect movement or alternatively if agricultural production eliminated agricultural margins?

Our estimates of connectivity dramatically improve if all agricultural margins function as perfectly natural habitat without any resistance due to pesticide loads. Results were largely consistent across metrics with most metrics showing 2 to 3 times more connectivity under the restored scenario compared to current conditions. Furthermore, the types of land cover included in the least-cost paths varied among the agricultural margin scenarios with agricultural margins being represented twenty-six times more in the natural margins scenario compared to current conditions or the scenario in which margins are converted to crops. The inclusion of the agricultural margins in the least-cost paths plus the shorter and more numerous paths that result from the natural margins scenario suggest that if agricultural margins could be enhanced this could greatly increase the permeability of the Central Valley form the perspective of pollinating insects. One of our unique contributions is that we consider the role that pesticides play in decreasing overall connectivity. Frequently, resistance surfaces are only related to land cover. We argue that it may be equally important, or perhaps more important, to consider actions occurring on or within a land cover (i.e. land use) within these

agroecosystems rather than just the actual land cover. For example, we found that under the low resistance scenario (in which pesticide use is the lowest) the number of unique paths was 10 times greater than the moderate resistance scenario and 18 times higher than the high resistance scenario. This suggest that reducing pesticide exposure has the potential to increase the number of highly connected paths.

Within the Central Valley of California there is a need to work with farmers to implement Integrated Pest Management programs that focus on reducing pesticide use in order to create corridors of lower pesticide exposure that allow pollinators to move freely and access habitat within the Central Valley, such as wildlife refuges, parks, open spaces, and agricultural fields where pollination services are highly valued. Government programs through federal and state agencies can help provide technical assistance and funding for farm conservation plans that include Integrated Pest Management goals. Farmers will change practices if they can tap into technical assistance and incentive programs. One example is Bee Better Certified, which is the first and only third-party certification program for pollinators. To gain Bee Better Certification status, farmers must have at least 1% of the farm landscape maintained as permanent pollinator habitat and an additional 4% in non-permanent habitat like cover crops, and must have a strong Integrated Pest Management program that protects habitat areas from pesticides. Bee Better Certification provides incentives to farmers because they and the companies that sell their products can promote their environmental credibility (for further discussion of policy issues see [3]). Local maps, such as those derived from this analysis (Fig 7), can help identify how habitat restoration and improvement around a focal area may result in greater connectivity for pollinators.

## Assumptions and future directions

Our framework provides a flexible starting point for thinking about how local actions, such as establishment of new hedgerows or other habitats, conservation easements, and changes in agricultural practices might influence overall landscape connectivity for pollinating insects. The framework is not intended to be accurate or predictive for any single pollinator species, but rather the goal has been to show broad-brush patterns in connectivity that may lead to discussions about where to locate conservation actions and spend conservation dollars. The framework is, however, quantitative and can be altered depending upon the assumptions being made. It can be used to test theories regarding land cover changes and changes to resistance within land cover types.

The biggest assumptions of our approach involve calculating resistance to movement, which is poorly understood and can be difficult to determine empirically for small animals including insects. We can hope that mark-recapture and genetic methods, while expensive and labor intensive [58], will continue to contribute important data on insect movement that could be incorporated into future iterations of our model. Another area of substantial uncertainty involves the pesticide records (and in particular the lack of data on insecticide treated seed), information on lethality, and how sublethal effects scale up to population, community and ecosystem consequences [59]. For example, our maps show large differences in resistance among the urban land cover types, which largely reflects uncertainty in urban areas due to a lack of reporting.

Our approach (like many similar efforts) using least-cost paths also relies on the assumption that cost is accumulated with each grid cell as an animal crosses the landscape. That might not be correct for animals with very long dispersal distances including larger butterflies, such as the monarch (*Danaus plexippus*). Use of least-cost paths and cost-distance for flying, pollinating insects should be restricted to species that are likely to move between adjacent 30-meter

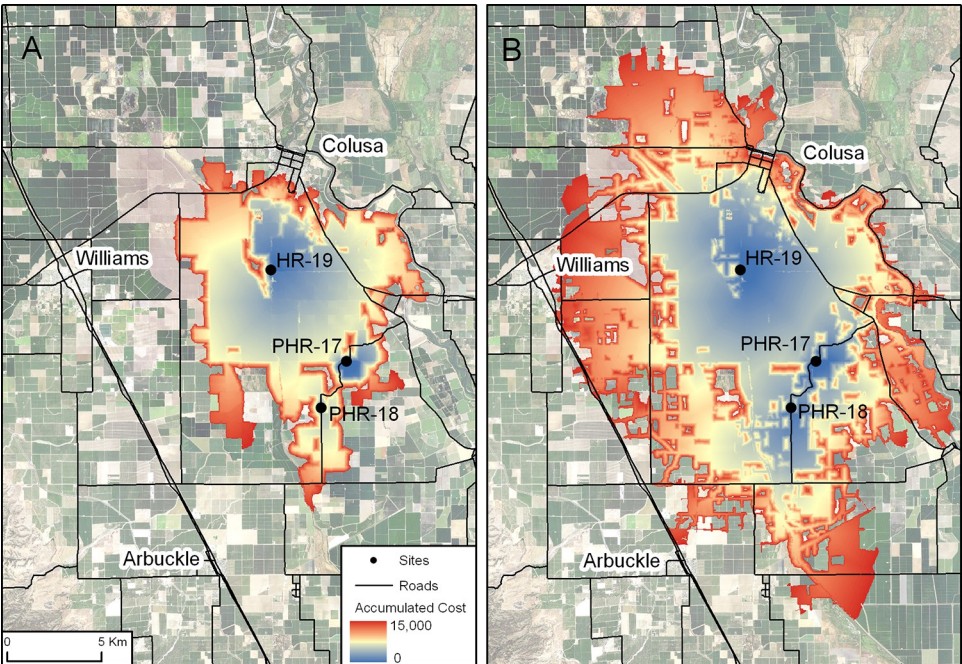

**Fig 7. Close-up image showing cumulative costs incurred as a function of distance moving outward from three source points (yellow dots) for A) the current conditions with moderate resistance and B) the scenario in which all agricultural margins are converted to natural habitat with moderate resistance for the other land cover classes.** Red colors indicate a low cost of movement (i.e. high probability of connectivity with the source points) and blue colors indicate a movement cost approaching 15,000 cost-distance units (number of cells traversed x cost of moving over the cell).

grid cells. It is worth noting in this context that the use of least-cost paths does not assume that a single animal will traverse that entire path. Rather, the approach represents general paths of least resistance across large landscapes that will likely take many generations, or will be used as continuous habitat even if no single individual ever completes the journey. In addition to incorporating new information on dispersal or pesticide lethality, our model is flexible and could be used to address a nearly endless list of additional questions. For example, one could investigate the effects of adding restored areas that replace agricultural fields or reducing pesticide loads on key crops. Such manipulations using modeling could be used to pose hypotheses that could then be tested with focused observations of specific species in the wild.

We assumed a static land cover layer and a static crop type layer. This is not a realistic assumption since many crops are rotated on frequent intervals. However, the availability of high-quality remotely sensed crop layers is limited, and typically remotely-sensed datasets take many years to produce. We used the 2014 LandIQ for all of our analyses. One possibility for future work would be to use the field geometry from this layer but provide annual crop type updates using the USDA Cropscape layer in order to allow for annual maps of crop types. As with many studies that use resistance surfaces, uncertainty in the particular resistance surface is a major reason why animals may fail to use the corridor as anticipated [60, 61]. Beyond the explicit modeling of temporal variation in crop types (and associated resistance), it would be useful to document and report which crops are rotated more frequently and which crops are relatively static. With our limited temporal window, this was not something that we considered part of the current project, but we highlight this temporal issue as an important issue for landscape ecologists in coming years.

Our analysis involves the use of three different resistance scenarios. Given the myriad of different body sizes, behaviors, and life history patterns of pollinating insects, it is unlikely that these three resistance surfaces capture the range of variability among pollinators. Resistance to movement is notoriously difficult to assess for a single species, let alone across a broad suite of taxa. In a 2012 review of studies that used resistance surfaces, Zeller et al. [62] found that the most common approach for estimating resistance was expert opinion and that the most commonly used data driven approaches are based on landscape genetics approaches that attempt to create an optimized resistance surface that best explains genetic differences among individuals or populations. Although our approach was initially based on a resistance surface derived using genetic data [38], we extrapolated all crop types and all other land cover types besides urban. Hence, our resistance surface likely functions more like a resistance surface derived using expert opinion and should be viewed cautiously. Further landscape genetics studies of larger suites of taxa would be tremendously beneficial for developing optimized resistance surfaces.

In summary, we view our analysis of cross-valley corridors as a macroscopic tool for assessing connectivity between natural lands on the eastern and western sides of the Sacramento Valley. Clearly more work is needed at finer spatial scales, especially for properties that might be completely isolated from the sides of the valley but could feasibly be connected either together or with nearby natural lands. Nevertheless, we hope that this effort is a useful stepping stone (pun intended) towards bigger and better models with more field-tested data. More generally, our results highlight the great and mostly untapped potential of marginal ag-adjacent lands for addressing the insect biodiversity crises and for sensible management of pollinators in complex anthropogenic landscapes.

## Acknowledgments

We thank Aimee Code, Angela Laws and Mace Vaughan (Xerces Society for Invertebrate Conservation) for insightful comments on an earlier version of the manuscript. MLF thanks the National Science Foundation for support (DEB-2114793). We also thank the many Xerces donors that have provided funding for this study.

## Author Contributions

**Conceptualization:** Thomas E. Dilts, Scott H. Black, Sarina J. Jepsen, Matthew L. Forister.

**Data curation:** Thomas E. Dilts, Sarah M. Hoyle, Emily A. May.

**Formal analysis:** Thomas E. Dilts, Matthew L. Forister.

**Funding acquisition:** Scott H. Black.

**Investigation:** Thomas E. Dilts, Matthew L. Forister.

**Methodology:** Thomas E. Dilts, Sarah M. Hoyle, Emily A. May.

**Project administration:** Matthew L. Forister.

**Resources:** Scott H. Black.

**Software:** Thomas E. Dilts.

**Supervision:** Sarina J. Jepsen, Matthew L. Forister.

**Validation:** Thomas E. Dilts.

**Visualization:** Thomas E. Dilts.

**Writing – original draft:** Thomas E. Dilts, Sarah M. Hoyle, Matthew L. Forister.

**Writing – review & editing:** Thomas E. Dilts, Scott H. Black, Sarah M. Hoyle, Sarina J. Jepsen, Emily A. May, Matthew L. Forister.

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
