## [Decision Letter · Decision Letter 0]

17 Jun 2022

PONE-D-22-09967Agricultural margins could enhance landscape connectivity for pollinating insects across the Central Valley of California, U.S.A.PLOS ONE

Dear Dr. Dilts,

Thank you for submitting your manuscript to PLOS ONE. After careful consideration, we feel that it has merit but does not fully meet PLOS ONE’s publication criteria as it currently stands. Therefore, we invite you to submit a revised version of the manuscript that addresses the points raised during the review process. The three reviewers have very positive comments about your paper, but suggested minor edits to improve it.

We look forward to receiving your revised manuscript.

Kind regards,

Alejandro Carlos Costamagna, Ph.D.

Academic Editor

PLOS ONE

Journal Requirements:

4. We note that Figures 2, 3 and 4 in your submission contain map/satellite images which may be copyrighted. All PLOS content is published under the Creative Commons Attribution License (CC BY 4.0), which means that the manuscript, images, and Supporting Information files will be freely available online, and any third party is permitted to access, download, copy, distribute, and use these materials in any way, even commercially, with proper attribution. For these reasons, we cannot publish previously copyrighted maps or satellite images created using proprietary data, such as Google software (Google Maps, Street View, and Earth). For more information, see our copyright guidelines: http://journals.plos.org/plosone/s/licenses-and-copyright.

a. You may seek permission from the original copyright holder of Figures 2, 3 and 4 to publish the content specifically under the CC BY 4.0 license.  

Reviewers' comments:

Reviewer's Responses to Questions

**Comments to the Author**

1. Is the manuscript technically sound, and do the data support the conclusions?

Reviewer #1: Yes

Reviewer #2: Yes

Reviewer #3: Yes

2. Has the statistical analysis been performed appropriately and rigorously? 

Reviewer #1: Yes

Reviewer #2: Yes

Reviewer #3: Yes

3. Have the authors made all data underlying the findings in their manuscript fully available?

Reviewer #1: Yes

Reviewer #2: Yes

Reviewer #3: Yes

4. Is the manuscript presented in an intelligible fashion and written in standard English?

Reviewer #1: Yes

Reviewer #2: Yes

Reviewer #3: Yes

5. Review Comments to the Author

Reviewer #1: This paper describes important progress in wildlife connectivity analysis. While many inputs remain speculative, the authors have made strides in applying analyses typically used on vertebrate species to an important new guild. When outputs such as these are combined with the more common connectivity assessments, we begin to get a clearer sense of how best to manage landscapes for future ecosystem viability. I recommend to the editors that this paper be published with minor revisions (below). The most important recommendation is to expand on how the LCP termini were selected, as these can have dramatic impacts on routes selected.

Line 183-192: Is this text the figure description or is it part of the main text?

Line 213: I think you mean multiplied, not divided.

Line 274-276: Why were these numbers chosen?

Line 279-282: It is not clear to me what this is for or really even means. Please explain.

Line 309-314: This probably belongs above where you first describe LCP.

Line 323-326: If I am reading this correctly, the combined land cover types only add up to 56.04% of the total Central Valley. Shouldn’t it be close to 100%. The same goes for the inner study area. Please explain or correct.

Line 532: “…focus ON insects…”

Line 542: One too many “Spencer et al.” I think.

Line 559-560: Your work is a great complement to Huber et al. It would be instructive to do the overlay you describe.

Reviewer #2: In this paper, the authors use publicly available data and expert opinion to estimate cost-distance surfaces (or resistance surfaces) and then model connectivity using least-cost paths across the central valley of California as a study site, with flying insect pollinators as model group. Additionally, they compare different potential land cover scenarios (with the status quo) to investigate the value of field margins for overall connectivity. They incorporate the pesticide intensity as a contributing factor to resistance – or landscape/matrix hostility – which is a very novel – and perhaps even ingenious – contribution to the concept of cost-distance modelling and how it relates to population/community stability and, in this case, sustainability of pollination services.

Overall, I am very much in support of the work. I like very much how the work is presented. I really like the use of such a great set of publicly available data in a practical and comprehensive way. The paper is well written and very polished. The authors clearly identify the limitations of the work without undermining the value and impact of the results. I also like very much the way they highlight future work and how the cost-distance models developed can be incorporated in future research and practical applications. I like how the introduction frames the work in Anthropocene, with the challenges to ecologists of food security and managing ecosystem functioning in increasingly fragmented and highly managed landscapes. The functional group they focus on is of high importance and is (always) topical; the study area as well. I like very much the inclusion of urban impervious and urban greenscapes as resistance surfaces – but on such a large scale these are probably overshadowed up by the larger scale processes. I love the interactive map linked in the text.

It is very interesting that the “resistance” in the matrix is also applied to the “habitat” in terms of pesticide intensity. Defining the quality of habitat in this way is excellent. And modelling the scenarios – or focusing on the value of margins – in both structural forming stepping stones, or more aptly, corridors, but also serving functionally as zones of refuge from pesticide application serves perfectly. This alone makes it an important contribution to the literature and how we think about habitat/matrix and landscape connectivity.

As a demonstration of practical use of available data, and an heuristic of the area wide impact of pesticide application and the value of margins for widescale connectivity, this paper serves as a very valuable contribution.

There are a number of minor, minor comments and suggestion that I have made in a tracked-changes version that I will upload. Please consider these comments as suggestion the authors might want to consider in order to sharpen and already very well-presented manuscript, but certainly do NOT make it unsuitable for publication without addressing. There were also some threads that I felt were raised in the results that could be tied up a bit more in the discussion if possible, which I will point out.

The points that I do annotate below should be addressed (or rebutted), but I will leave other tracked-changes and embedded comments up to the discretion of the authors.

Ln80 where you are listing the counties of southern Sacramento valley, these are a bit meaningless for readers who don’t have a working knowledge of geography of the area. Perhaps include reference to a map here, if they are important enough to note by name. Otherwise, do you really need to name them?

Ln198 what are you trying to say here? There is a bit of a word jumble going on.

Ln199-200 extra parenthesis on one of these lines (or an opening bracket missing)

Ln257 consider rewording. Not immediately clear what is inferred here. Perhaps it just needs a ‘but’ or use ‘varies’ in place of ‘changes’. Reads a little odd as is, at least to me.

Ln268 secondly? Perhaps

Figure 2 Stockton labelled twice. One should be Fresno? Figure 3 has it correct, it would seem.

Ln317 should it be ‘are larger’ rather than ‘is larger’?

Ln345 do you mention anywhere above that Central Valley is divided into the Sacramento Valley in the north and San Joaquin Valley in the south? If you haven’t then make sure you do. If I missed it, I do apologize.

Fig5 just be careful with font size across all panels. Also it is a little distracting that you have used a different scale – and therefore have contrasting numbers of zeros – across panels.

Ln475 should this be Fig6 you are referencing?

Lns 538, 543 just check that you don’t have a formatting/editing error. The latter seems to have too many Spencer et alia

Reviewer #3: Overall, this paper is very clear, concise and provides a useful way of including agro-chemicals into connectivity mapping. This is a highly in-demand topic, especially in the context of wild pollinators in an agro-ecosystem.

Here I provide a very brief summary of the paper in my own words:

The authors hypothesize that marginal habitat (including habitats such as agricultural margins and roadside ditches) is an important factor for wild pollinator connectivity. To test this, the authors simulated least-cost paths across the Central Valley of California in a 3x3 design. The design consisted of low, medium, or high resistance maps in combination with no marginal habitat, status-quo marginal habitat, and restored to “natural” marginal habitat. Resistance maps were created using land cover, referenced literature, and expert opinion. Additionally, they incorporated agro-chemical use (mostly pesticides) into these resistance values. The authors hypothesize that restoring marginal habitat quality will improve native pollinator connectivity and by extension, aid conservation efforts for pollinator abundance, diversity, ecosystem services.

Overall, I thought the paper was very good. The authors did a good job of contextualizing the work locally, particularly how the methods compare to other connectivity projects taking place in California. However, I would strongly encourage a more thorough integration into the ecology literature, as there is an abundance of work, particularly in Europe, investigating the importance of roadside ditches, field margins, and hedgerows on pollinators and other insects. I would also encourage the authors to more clearly state their a priori expectations and relate them to ecology literature.

Throughout the paper:

Consider rounding large numbers to a relevant number of significant digits.

Many specific areas of California are mentioned that do not appear on maps in the provided figures. Please consider adding these areas on a map or consider including an additional map to visualize these areas. When considering a broad audience, the authors should not expect most readers to be so familiar California geography.

Abstract

L 17 - 20: There are a lot of ideas here. I recommend finding a simpler, more concise way to phrase this sentence or break it up into smaller sentences.

Introduction

L 43 – 45, 61 – 68, 71 -75: Additional citations are needed here.

L 98 – 99: Further explanation around “if a crop is typically treated with a high chemical LD50,

” is needed.

L 96: Consider including a more formal definition of “resistance”.

L 101-106: Might the thoughts (beginning at “We acknowledge that”) be better suited to the Discussion section?

Materials and Methods:

Study Area

L 121 – 127: Can you provide a source for this or is this direct observation?

Land Use Data

L 138: This sentence contains a lot of information and is difficult to read. The phrasing should be simplified and broken down into separate parts. For example, “We created a dataset (from sources detailed below) that incorporates land cover and pesticide application rates across the Central Valley.”

L 142 - 143: Unnecessary comma after “ground data”. Consider turning it into a compound sentence or deleting the second phrase.

Resistance surfaces

L 165: “We compiled data on moderately and highly bee-toxic pesticide” Did you exclude any pesticides? If so, what criteria did you use? Is there a chance that an excluded pesticide could have an effect on bees? Were neonicotinoids used at all?

L 180, 193, 197, 239: Inconsistent font type

L 206 – 209: Your meaning here is unclear. You say “(taking those values from the lowest, middle and highest values across the three years)”, which is easy to interpret in different ways. Please add some clarification. As I understand it, the per-acre toxic load of each crop type varies across years. And you calculated the per-acre toxic load for each year for each crop type, resulting in three different values per crop type. These values were used to create the low, medium, and high resistance versions of the map during the study year.

L 211: Many readers will not be familiar with specific agrochemicals. A brief explanation of Spinosad would be very helpful, for example “Spinosad (a commonly used insecticide in the US)”

L 236: Consider adding a comma before “rounding” to separate phrases

L 287 – 291: Please add clarification. I’m not sure what you’re saying here. Is this conceptually similar to current density?

Results

Discussion

L 477 – 513: More citations are needed here.

L 532: “a focus on insects and their habitat”

L 536: What is “(page 53)” referencing here? There is no page 53 of this document

L 598: Acronym “IPM” should be defined on line 593 where you introduce the term Integrated Pest Management

L 599: What is the farmer’s incentive for earning Bee Better Certification?

L 668: “likely functions”

6. PLOS authors have the option to publish the peer review history of their article (what does this mean?). If published, this will include your full peer review and any attached files.

Reviewer #1: No

Reviewer #2: No

Reviewer #3: No

---

## [Author Response · Author response to Decision Letter 0]

20 Sep 2022

Responses are also contained in the cover letter with original comments in black and responses italicized blue font.

Editor’s comments

We have reviewed the style requirements and have followed them in this set of revisions.

2. In your Data Availability statement, you have not specified where the minimal data set underlying the results described in your manuscript can be found.

We have made our data available on Dryad (doi:10.5061/dryad.pc866t1s4).

3. We note that you have stated that you will provide repository information for your data at acceptance.

We have made our data available on Dryad (doi:10.5061/dryad.pc866t1s4). Datasets posted on Dryad include the compiled pesticides data, resistance values for each land cover type, maps of the study areas (both Central Valley and close-up), the landcover map used in the analysis, resistance surfaces for all nine combinations of resistance + agricultural margin scenarios, results of the least-cost path analysis for both the Central Valley and close-up study areas, and the summary statistics derived from the least-cost analysis.

4. We note that Figures 2, 3 and 4 in your submission contain map/satellite images which may be copyrighted.

Thank you. We have revised Figures 2, 3, and 7. Figure 2 never contained satellite imagery, but Reviewer 2 noted that Stockton was accidentally labeled twice, which we have now corrected. We have also checked this figure in Coblis to ensure that it is okay for color blindness and it appears to be readable for readers with both protanopia and deuteranopia despite the fact that it uses reds and greens - https://www.color-blindness.com/coblis-color-blindness-simulator/. We have removed the underlying basemaps in Figure 3 since they were not legible at this scale. We replaced the basemaps in Figure 7 with freely available National Agriculture Imagery Program data for 2020 in California - https://data.ca.gov/dataset/naip-2020-natural-color-california.

5. Please review your reference list to ensure that it is complete and correct.

Reviewer #1

Line 183-192: Is this text the figure description or is it part of the main text?

These lines are all part of the figure 1 caption.

Line 213: I think you mean multiplied, not divided.

Correct. Thank you for pointing out that mistake. We have corrected it.

Line 274-276: Why were these numbers chosen?

Changed from “Corridor mapping was performed using least-cost paths [35] between 205 source points on the western perimeter of the study area and 205 points on the eastern perimeter of the study area. The average distance between a point and its nearest neighbor was about 5 km.” to “Corridor mapping was performed using least-cost paths [35] connecting source points on the western and eastern perimeter of the study area. The average distance between a point and its nearest neighbor was about 5 km, resulting in 205 source points along the western perimeter and 205 source points along the eastern perimeter.” This reorganization emphasizes that it was the point spacing that came first rather than the number of points.

Line 279-282: It is not clear to me what this is for or really even means. Please explain.

We probably were not clear enough in this section. The main point is that in order to account for variation due to the location of the start and end points we needed to have an iterative process that accounts for random variation in start and end points. We have changed the text from "We repeated this process for nineteen iterations using randomly-drawn points from 18 km2 polygons along the western perimeter as source points and randomly-drawn points from 18 km2 polygons along the eastern perimeter as destination points." to "To account for variability in start and end points we repeated the process totaling 20 iterations using random points within a 5 km buffer of original source and end points."

Line 309-314: This probably belongs above where you first describe LCP.

We have moved the following text up to the beginning of the section titled Analyses: connectivity models:

“Least-cost paths were calculated using the Cost Distance and Cost Path as Polyline tools in ArcGIS Spatial Analyst version 10.7.1. Cost-distance is calculated using Dijkstra's algorithm, which is an iterative algorithm that finds the shortest distance between all nodes and a source node in a graph, and the least-cost path is identified by taking the accumulated cost (resistance) traversed along the least-cost path.”

Line 323-326: If I am reading this correctly, the combined land cover types only add up to 56.04% of the total Central Valley. Shouldn’t it be close to 100%. The same goes for the inner study area. Please explain or correct.

We didn’t originally report grassland in this paragraph which makes up everything else. We have changed the text from “Combined crops accounted for 35.26% of the larger Central Valley and 65.46% of the inner study area.” to read “Crops accounted for 35.26% of the larger Central Valley that includes both the inner and outer study areas (Figure 2). Within the more intensively cropped inner study area crops accounted for 65.46% of the total area.” to refer the reader to Figure 2 for the inner and outer study area. We have also added the sentence “The remaining land cover was grassland constituting 37.81% of the larger study area or 11.50% of the inner study area.”

Line 532: “…focus ON insects…”

Thank you. Change made.

Line 542: One too many “Spencer et al.” I think.

Thank you. Change made.

Line 559-560: Your work is a great complement to Huber et al. It would be instructive to do the overlay you describe.

We can't disagree! But we do not think the current paper needs any more figures or analyses; we have added text to the relevant sentence indicating that the analysis in question is "beyond the scope of the current project.")

Reviewer #2

Ln80 where you are listing the counties of southern Sacramento valley, these are a bit meaningless for readers who don’t have a working knowledge of geography of the area. Perhaps include reference to a map here, if they are important enough to note by name. Otherwise, do you really need to name them?

We have removed the reference to the five counties and added a polygon on Figure 2 to show the location of the close-up study area with labeling.

Lm 170 - You could be more taxonomically specific.

Change made.

Ln198 what are you trying to say here? There is a bit of a word jumble going on.

Changed to read “We did not include the ‘structural pest control’ category because it is not known which of these application methods (i.e. applications in and around buildings to control damage from termites and other structural pests) poses a significant risk to pollinators.”

Ln199-200 extra parenthesis on one of these lines (or an opening bracket missing).

Thanks. See previous comment.

Ln 213 - The total area, I assume.

Thanks. Added (total area) after 453,592,370. 

Ln257 consider rewording. Not immediately clear what is inferred here. Perhaps it just needs a ‘but’ or use ‘varies’ in place of ‘changes’. Reads a little odd as is, at least to me.

Good point. We changed “changes” to “varied” in this sentence.

Ln268 secondly? Perhaps

Change made.

Figure 2 Stockton labelled twice. One should be Fresno? Figure 3 has it correct, it would seem.

Thank you for catching this error. We have made this correction. We have also checked this figure in Coblis to ensure that it is okay for color blindness and it appears to be readable for readers with both protanopia and deuteranopia despite the fact that it uses reds and greens - https://www.color-blindness.com/coblis-color-blindness-simulator/.

Ln 316 - Subheadings to guide the results, possibly.

Have added the following subheadings: Land cover analysis, Resistance values.

I guess, overall, I felt the results start off in an underwhelming fashion. The resistance values are more interesting than the general accounting type of percentage cover of different land uses. It might be a bit more engaging to start by talking about the resistance values directly, first off, and then fold-in the percentage cover of particularly/relevant cover types.

Think about. Once we know the resistance values we might be more engaged with the relative abundance of each.

We appreciate this perspective, and we did experiment with different arrangements of the results, but we found that it was hard to get back to those basic descriptive stats if we didn't have them right upfront. We have, however, added a new section heading ("Land cover analysis") to the beginning of the results section, which we hope will at least make it easier for the reader to navigate.

Ln317 should it be ‘are larger’ rather than ‘is larger’?

Change made.

Ln 318 Any one particular crop?

Changed from “any one individual crop” to “most crops”.

Ln 333 – 336 - Awkward sentence

Split into two sentences.

Ln345 do you mention anywhere above that Central Valley is divided into the Sacramento Valley in the north and San Joaquin Valley in the south? If you haven’t then make sure you do. If I missed it, I do apologize.

Added a brief reference in the study area section of the methods. In the results section we added “(southernmost 1/3rd of the study area).”

Ln 349 – 351 - As I was reading through this the first time, there were a number of points in the results section that I thought, ‘it will be interesting to see how this is addressed in the discussion’

And then when I got to the discussion, I saw that you went in an even more interesting and broad scope. Without discussing the results perhaps you can just comment on the significance or limitations of some the results I highlight. Or include a paragraph in the discussion that touches on these points. I think it will help to just round off or tie up any potential loose ends.

We added the sentence “For example, our maps show large differences in resistance among the urban land cover types, which largely reflects uncertainty in urban areas due to a lack of reporting.” to the section discussing uncertainty in pesticide reporting in the discussion.

Fig5 just be careful with font size across all panels. Also it is a little distracting that you have used a different scale – and therefore have contrasting numbers of zeros – across panels.

We have revised Figure 5 to ensure that the font sizes are all consistent. Unfortunately, it probably won’t be possible to have a unified scale across all graphs given the differences between the metrics and the differences among the Central Valley and the close-up study areas. We have added the sentence “Note the differences in the y-axis among graphs.” to the figure caption.

Ln 362 - A small number? Pairing few with few

We added “(around 14 in the current margins scenario)” to qualify this statement and omitted “a few”.

Ln 369 - How is this picked up in the discussion?

We added the sentences “For example, we found that under the low resistance scenario (in which pesticide use is the lowest) the number of unique paths was 10 times greater than the moderate resistance scenario and 18 time higher than the high resistance scenario. This suggest that reducing pesticide exposure has the potential to increase the number of highly connected paths.” to the discussion.

Ln 370 - Careful with this, or is it ‘that’?

By which I mean, I think that throughout you need to be more careful to put a comma before ‘which’ because of the way you are using it as a conjunction, which forms a subclause. (As opposed to ‘that’, which doesn’t.)

We did a find and replace to add commas in front of “which” where appropriate.

Ln 374 - ‘and’? ‘,where’? ‘in that’?

Changed “and” to “in that”.

Ln 412 – 414 - This is a little bit of complicated sentence. You might want to round it off, or compliment it, with a plan language explanation of what it indicates, or what to look for when trying to interpret results. Like you’ve done in ln408

We have changed the text from:

"The average number of convergent paths indicates the average number of least-cost paths traversing a single cell within the inner area considering only those cells that are located within paths."

to:

"The average number of convergent paths may suggest a funneling effect in which paths converge into a handful of low-resistance areas, and is measured by measuring the number of times that least-cost paths traverse a single cell within the inner study area."

Ln 422 – Lower?

Change made. Thank you.

Ln 428 - How is this addressed in the discussion?

We have added the following sentences to the discussion: "Furthermore, the types of land cover included in the least-cost paths varied among the agricultural margin scenarios with agricultural margins being represented twenty-six times more in the natural margins scenario compared to current conditions or the scenario in which margins are converted to crops. The inclusion of the agricultural margins in the least-cost paths plus the shorter and more numerous paths that result from natural margins scenario suggests that if agricultural margins could be enhanced this could greatly increase the permeability of the Central Valley form the perspective of pollinating insects."

Ln475 should this be Fig6 you are referencing?

Change made. Thank you.

Ln 495 – 497 - Food for thought, though, why aren’t they more popular with growers? Where are we missing with the messaging?

That is an interesting point, and we don't have the answer. At the risk of baseless speculation, we believe we should probably leave this issue without further elaboration in the current text, but it is something we will think about for future work.

Ln 507 - Stronger word that move perhaps.

We changed this to “shift their distributions”.

Ln 511 – 513 - This is the more important benefit to highlight first off, I think. The benefit of connectivity in the immediate future is more stable pollinator communities and sustainability pollination services.

We agree strongly with you and have added the sentence “The benefits of connectivity include more stable pollinator communities and sustainable pollination services.” Thank you for the suggestion.

Ln 525 - Land cover types?

Changed “land” to “land cover types”.

Ln 525 - Are largely

Changed “largely are” to “are largely”.

Lns 538, 543 just check that you don’t have a formatting/editing error. The latter seems to have too many Spencer et alia

Thank you. Somehow Spencer et al. was duplicated. We have also combined the two sentences to read “There are several notable differences in the methodology between our study and Spencer et al. [42], which only considered large connectivity among protected habitat blocks.”

Ln 542 - Capital letter of Corridor or “corridor 6” in parentheses?

We changed “corridor” to “Corridor”.

Ln 546 - Suggestion only

We changed “In contrast, our study” to “Our study, in contrast,”.

Ln 552 - Reconsider wording here

Changed from:

"Another relevant effort is from Huber et al. [43], who created corridors for eight umbrella species (tule elk, bobcat, giant garter snake, pronghorn, western yellow-billed cuckoo, riparian forest, San Joaquin kit fox, Swainson’s hawk) across the Central Valley."

to "Huber et al. [43] created corridors for eight umbrella species (tule elk, bobcat, giant garter snake, pronghorn, western yellow-billed cuckoo, riparian forest, San Joaquin kit fox, Swainson’s hawk) across the Central Valley."

Ln 562 – 564 - Reconsider how you start this paragraph too

Changed from:

"In a study that examined the spatial overlap of different corridor mapping approaches in California, Choe et al. [44] examined the maps from three different studies using three different connectivity modeling approaches."

to

"Choe et al. [44] examined the maps from three different studies using three different connectivity modeling approaches across the state of California, including the Central Valley."

Ln 566 - Although? In addition to? Beyond just?

We omitted “Although.”

Ln 566 – 572 - Not sure about this section

Does each study refer to refs 45-47 or all studies mention above (42-47) compared to your own present work?

Either way it isn’t really tying together or emphasizing the value of your work – more so feels like it is undermining the value of your contribution.

Omit or sum up a bit more conclusively (or emphatically).

We agree that as-written this section undersells our work quite a bit and probably creates confusion for the reader. We have modified and expanded this section. We also realized that the Choe et al. (2020) paper included a fourth type of corridor – metacorridors. The authors of that paper apply their metacorridor approach in their paper and compare it with the other three approaches. Importantly, none of the approaches in Choe et al. (2020) highlight the Central Valley as being important and none of the studies examine pollinator connectivity.

We changed the text from:

"The three different approaches included Network Flow Analysis [45], Land Facets analysis [46], and Omniscape [47]. They found that there was relatively little overlap between these three sets of corridors. Although each study used a different methodological approach they also all differed in objectives, the species under consideration, and the type of resistance surfaces used in models. Our approach represents yet another set of considerations, but it is unique from these other studies in that it focuses on the agricultural lands of the Central Valley, it incorporates pesticide application rates into a measure of landscape resistance, and it explicitly considers how agricultural margins potentially facilitate connectivity through a complex and heterogeneous landscape."

to

"The four different approaches included the Metacorridor approach [44], Network Flow Analysis [45], Land Facets analysis [46], and Omniscape [47]. They found that there was relatively little overlap between the four sets of corridors. Each study used a different methodological approach, differed in their research objectives, and differed in the type of resistance surfaces used in models. Two approaches used focal plant species (Metacorridors and Network Flow Analysis) whereas the other two connected areas of low human impact (Land Facet Corridors and Omniscape). Importantly, none of the four studies reviewed by Choe et al. [44] include insect pollinators as a focal species, and maps from all four studies (as shown in Choe et al. [44], Figure 1) show a lack of corridors in the Central Valley. Corridor maps that omit highly developed landscapes may give the impression that highly modified landscapes are not restorable or valuable for connectivity. In contrast, changing economic and social conditions may present opportunities for habitat enhancement in highly modified landscapes that even exceed those of natural landscapes. For example, adoption of agricultural practices that use no or very low levels of pesticides may greatly enhance connectivity. Losses of farmland due to drought may also present opportunities for habitat enhancement by incorporating plantings of low-water plants that provide resources for pollinators.”

Ln 574 - High quality? Fine detail?

Changed “good quality” to “high resolution”.

Ln 600 – 604 - More food for thought – how can you engage with growers and govt agencies to achieve these goals?

Instead of adding additional text to the Discussion, which we believe could veer off topic, we have added a citation to our previous work that included more detailed discussions of policy and engagement with growers.

Ln 632 – 634 - Although, even there, butterflies do tend to follow margins – or at least be associated with margins and are more abundance in agricultural landscape with greater edge density of seminatural areas.

We added “such as the monarch (Danaus plexippus).” to this sentence. We believe that even for flying insects most interact with their environment in more of a lattice-model in which movement occurs among adjacent grid cells compared to a tele-connectivity model in which movement largely skips over adjacent cells. This is an area that could benefit from further research.

Ln 640 – 645 - It is great that you are forward thinking with utility of this model. Perhaps you could make more of that upfront – e.g., in the abstract.

I mean, you do that already. But emphasize it more like a selling point or branding of your ‘product’ or framework. 

Just a selling point to consider.

We have added the sentence “Our modeling approach is flexible and can be used to address a wide range of questions regarding both changes in land cover as well as changes in pesticide application rates.” to the abstract.

Ln 660 - I would put the ‘however’ here after ‘insects’

We split this sentence into two since it was long.

Reviewer #3

L 17 - 20: There are a lot of ideas here. I recommend finding a simpler, more concise way to phrase this sentence or break it up into smaller sentences.

Changed from “One of the defining features of the Anthropocene is eroding ecosystem services as a function of decreases in biodiversity and overall reductions in the abundance of once-common organisms, including many insects that play innumerable roles in natural communities and agricultural systems that support human society.” to “One of the defining features of the Anthropocene is eroding ecosystem services, decreases in biodiversity, and overall reductions in the abundance of once-common organisms, including many insects that play innumerable roles in natural communities and agricultural systems that support human society.”

Introduction

L 43 – 45, 61 – 68, 71 -75: Additional citations are needed here.

Citations have been added to these sections.

L 96: Consider including a more formal definition of “resistance”.

We have added the following sentence “We can depict resistance to movement on maps referred to as “resistance surfaces” that quantitatively express the relative difficulty of moving across one grid cell relative to another.” and have moved the Zeller et al (2012) paper up in the reference list.

L 98 – 99: Further explanation around “if a crop is typically treated with a high chemical LD50,

” is needed.

Added “(i.e. more lethal dose),”.

L 101-106: Might the thoughts (beginning at “We acknowledge that”) be better suited to the Discussion section?

We understand why it might seem like Discussion material, but we feel strongly that this caveat about "assumptions and simplifications" needs to be read before the reader encounters all of the methods (and the associated assumptions). Thus we prefer to keep it where it is, but if the reviewer or editor feels strongly that we are wrong, we would be willing to change it.

L 121 – 127: Can you provide a source for this or is this direct observation?

We added Bertoldi GL 1989, Ground-water resources of the Central Valley of California: U.S. Geological Survey Open-File Report 89–251, 2 p

L 138: This sentence contains a lot of information and is difficult to read. The phrasing should be simplified and broken down into separate parts. For example, “We created a dataset (from sources detailed below) that incorporates land cover and pesticide application rates across the Central Valley.

We changed the text from “We combined a number of land cover products into a single unified land cover dataset and then incorporated pesticide records collected by the California Department of Pesticide Regulation to generate maps of pesticide application rates across the Central Valley.” to “First, we combined a number of land cover products (LandIQ, C-CAP, NLCD) into a single unified land cover dataset. Next we then incorporated pesticide records collected by the California Department of Pesticide Regulation to generate maps of pesticide application rates across the Central Valley.”

L 142 - 143: Unnecessary comma after “ground data”. Consider turning it into a compound sentence or deleting the second phrase.

We removed the comma and split the sentence into two.

L 165: “We compiled data on moderately and highly bee-toxic pesticide” Did you exclude any pesticides? If so, what criteria did you use? Is there a chance that an excluded pesticide could have an effect on bees? Were neonicotinoids used at all?

We did not exclude any pesticides from our analysis, but did rely strongly on DPR’s Pesticide Use Reports. In some cases, DPR does not require reporting. For example, treated seed is not reported in the DPR data. We acknowledge that most likely our estimates are underestimates due to these omissions in reporting. We have included the major chemicals for the crops with the most toxic loads in our supplementary data which is available on Dryad. In the text we highlight the lack of data on treated seed as well as the lack of reporting in urban areas as potential data gaps.

“DPR’s Pesticide Use Reports do not include information on the planting of pesticide treated seed, which could lead to the underestimation of the toxic load for crops that use insecticide-treated seed including alfalfa, corn, cotton, rice, squash, sunflowers and wheat [37].” Page 9

“Urban pesticide use was estimated from DPR’s Pesticide Use Reports for non-agricultural use categories in the 20 Central Valley counties. Since many urban pesticide applications are not required to be reported (including from residents applying pesticides around their homes), this method provides a potentially dramatic underestimate of actual rates of pesticide application to these landscapes. We did not include the ‘structural pest control’ category because it is not known which of these application methods (i.e. applications in and around buildings to control damage from termites and other structural pests) poses a significant risk to pollinators. Reported urban applications (in pounds) for each pesticide were converted to a per-acre toxic load using the same methodology as for crops described above.” Page 10

L 206 – 209: Your meaning here is unclear. You say “(taking those values from the lowest, middle and highest values across the three years)”, which is easy to interpret in different ways. Please add some clarification. As I understand it, the per-acre toxic load of each crop type varies across years. And you calculated the per-acre toxic load for each year for each crop type, resulting in three different values per crop type. These values were used to create the low, medium, and high resistance versions of the map during the study year.

Changed from

“Although our model is temporally static, we wished to incorporate some of that variation, thus we used the pesticide data in hand (for 2014, 2015 and 2016) to set low, medium and high values for each crop type (taking those values from the lowest, middle and highest values across the three years).”

to

“Although our model is temporally static, we wished to incorporate variation among years. To incorporate interannual variation in pesticide application rates we used each of the three years for which we had data (2014, 2015, and 2016) and assigned low, medium, and high pesticide application rates according the lowest, median, and highest within the three year period.”

L 211: Many readers will not be familiar with specific agrochemicals. A brief explanation of Spinosad would be very helpful, for example “Spinosad (a commonly used insecticide in the US)”

Change made.

L 236: Consider adding a comma before “rounding” to separate phrases

Change made.

L 287 – 291: Please add clarification. I’m not sure what you’re saying here. Is this conceptually similar to current density?

We have removed this section because we don’t display these results in any of the Figures and for clarity. Rather we only show the results of the least-cost path lines themselves and never polygons.

L 477 – 513: More citations are needed here.

Citations have been added to this section.

L 532: “a focus on insects and their habitat”

Change made.

L 536: What is “(page 53)” referencing here? There is no page 53 of this document

Changed to read “page 53 in the California Essential Habitat Connectivity Project”.

L 598: Acronym “IPM” should be defined on line 593 where you introduce the term Integrated Pest Management

Change made.

L 599: What is the farmer’s incentive for earning Bee Better Certification?

We have added the sentence “Bee Better Certification provides incentives to farmers because they and the companies that sell their products can promote their environmental credibility.”

There are several ways in which farmers can benefit from BBC certification. 1) several companies have stepped up to provide direct support to their supply chain farmers to comply with BBC certification by funding restoration and IPM development. 2) Farmers can get a premium for the products they sell. 3) farmers can use this to promote their products as environmentally friendly (example would be at a farmers market). 4) Companies can use products from their supply chain to promote their environmental credibility which can help sales. 5) Some stores like Walmart are requiring produce meet environmental standards and BBC certifications (along with organic certification and others) meet these standards.

L 668: “likely functions”

Change made.

---

## [Decision Letter · Decision Letter 1]

26 Oct 2022

Agricultural margins could enhance landscape connectivity for pollinating insects across the Central Valley of California, U.S.A.

PONE-D-22-09967R1

Dear Dr. Dilts,

We’re pleased to inform you that your manuscript has been judged scientifically suitable for publication and will be formally accepted for publication once it meets all outstanding technical requirements.  Congratulations.  Really interesting study!  

I found a couple of minor edits that need to be corrected.  Before forwarding the final version for publication, please be sure these are corrected. 

L73 - There should be a comma before 'which'L614 - 'form' should be 'from'L670-672 - Wording here is awkward & the meaning unclear.  Do you mean: With respect to pollinating insects, use of least cost ... should be restricted to ... ?PLOS ONE does not use a copy editor and the onus is on the authors to ensure typos/grammatical errors etc. are eliminated.  In addition to the two minor errors I already noted, I found a number of inconsistencies and errors in the references just on a quick look through, e.g. missing or extra spaces, inconsistent use of comma vs. colon, etc.  Please carefully read through the entire manuscript to uncover any remaining errors.  If possible, I'd suggest having someone unfamiliar with the work to read through closely as they would be more likely to spot errors and inconsistencies.  

Kind regards,

Dr. Janice L. Bossart

Academic Editor

PLOS ONE

Additional Editor Comments (optional):

Reviewers' comments:

Reviewer's Responses to Questions

**Comments to the Author**

1. If the authors have adequately addressed your comments raised in a previous round of review and you feel that this manuscript is now acceptable for publication, you may indicate that here to bypass the “Comments to the Author” section, enter your conflict of interest statement in the “Confidential to Editor” section, and submit your "Accept" recommendation.

Reviewer #1: All comments have been addressed

Reviewer #3: All comments have been addressed

2. Is the manuscript technically sound, and do the data support the conclusions?

Reviewer #1: Yes

Reviewer #3: Yes

3. Has the statistical analysis been performed appropriately and rigorously? 

Reviewer #1: Yes

Reviewer #3: Yes

4. Have the authors made all data underlying the findings in their manuscript fully available?

Reviewer #1: Yes

Reviewer #3: Yes

5. Is the manuscript presented in an intelligible fashion and written in standard English?

Reviewer #1: Yes

Reviewer #3: Yes

6. Review Comments to the Author

Reviewer #1: (No Response)

Reviewer #3: (No Response)

7. PLOS authors have the option to publish the peer review history of their article (what does this mean?). If published, this will include your full peer review and any attached files.

Reviewer #1: No

Reviewer #3: No

---

## [Editor Report · Acceptance letter]

26 Jan 2023

PONE-D-22-09967R1 

Agricultural margins could enhance landscape connectivity for pollinating insects across the Central Valley of California, U.S.A. 

Dear Dr. Dilts:

I'm pleased to inform you that your manuscript has been deemed suitable for publication in PLOS ONE. Congratulations! Your manuscript is now with our production department. 

Kind regards, 

on behalf of

Dr. Janice L. Bossart 

Academic Editor

PLOS ONE